# Selective Inhibition of Soluble Tumor Necrosis Factor Alters the Neuroinflammatory Response following Moderate Spinal Cord Injury in Mice

**DOI:** 10.3390/biology12060845

**Published:** 2023-06-12

**Authors:** Minna Christiansen Lund, Ditte Gry Ellman, Pernille Vinther Nielsen, Stefano Raffaele, Marta Fumagalli, Raphael Guzman, Matilda Degn, Roberta Brambilla, Morten Meyer, Bettina Hjelm Clausen, Kate Lykke Lambertsen

**Affiliations:** 1Department of Neurobiology Research, Institute of Molecular Medicine, University of Southern Denmark, 5000 Odense, Denmark; minnacl@hotmail.com (M.C.L.); dellman@health.sdu.dk (D.G.E.); pvnielsen@health.sdu.dk (P.V.N.); stefano.raffaele@unimi.it (S.R.); rbrambilla@med.miami.edu (R.B.); mmeyer@health.sdu.dk (M.M.); bclausen@health.sdu.dk (B.H.C.); 2Department of Neurology, Odense University Hospital, 5000 Odense, Denmark; 3Department of Pharmacological and Biomolecular Sciences, Università degli Studi di Milano, 20133 Milan, Italy; marta.fumagalli@unimi.it; 4Department of Biomedicine, University of Basel, 4031 Basel, Switzerland; raphael.guzman@usb.ch; 5Department of Paediatrics and Adolescent Medicine, Rigshospitalet, 2100 Copenhagen, Denmark; matildadegn@gmail.com; 6The Miami Project to Cure Paralysis, Department of Neurological Surgery, University of Miami Miller School of Medicine, Miami, FL 33136, USA; 7Brain Research Inter-Disciplinary Guided Excellence (BRIDGE), Department of Clinical Research, 5000 Odense, Denmark; 8Neuroscience Program, University of Miami Miller School of Medicine, Miami, FL 33136, USA

**Keywords:** XPro1595, neuroinflammation, microglia, CNS trauma, leukocyte infiltration

## Abstract

**Simple Summary:**

Blocking detrimental neuroinflammation and boosting a pro-regenerative environment are promising new therapeutic approaches in spinal cord injury (SCI). Selective blocking of the soluble (sol) form of the cytokine tumor necrosis factor (TNF) while maintaining the membrane-bound form of TNF (tmTNF) has shown beneficial effects in pre-clinical models of SCI. In the present study, we investigated the effects of selective solTNF inhibition on the spatio-temporal inflammatory response after SCI in mice. We found that the blocking of solTNF alleviated the pro-inflammatory response, altered microglial responses, increased myelin preservation, and improved functional outcomes. Altogether, this demonstrates that selective inhibition of solTNF holds translational potential after SCI.

**Abstract:**

Clinical and animal model studies have implicated inflammation and glial and peripheral immune cell responses in the pathophysiology of spinal cord injury (SCI). A key player in the inflammatory response after SCI is the pleiotropic cytokine tumor necrosis factor (TNF), which exists both in both a transmembrane (tmTNF) and a soluble (solTNF) form. In the present study, we extend our previous findings of a therapeutic effect of topically blocking solTNF signaling after SCI for three consecutive days on lesion size and functional outcome to study the effect on spatio-temporal changes in the inflammatory response after SCI in mice treated with the selective solTNF inhibitor XPro1595 and compared to saline-treated mice. We found that despite comparable TNF and TNF receptor levels between XPro1595- and saline-treated mice, XPro1595 transiently decreased pro-inflammatory interleukin (IL)-1β and IL-6 levels and increased pro-regenerative IL-10 levels in the acute phase after SCI. This was complemented by a decrease in the number of infiltrated leukocytes (macrophages and neutrophils) in the lesioned area of the spinal cord and an increase in the number of microglia in the peri-lesion area 14 days after SCI, followed by a decrease in microglial activation in the peri-lesion area 21 days after SCI. This translated into increased myelin preservation and improved functional outcomes in XPro1595-treated mice 35 days after SCI. Collectively, our data suggest that selective targeting of solTNF time-dependently modulates the neuroinflammatory response by favoring a pro-regenerative environment in the lesioned spinal cord, leading to improved functional outcomes.

## 1. Introduction

Tumor necrosis factor (TNF) is a potent immunomodulatory cytokine with crucial functions in initiating and regulating inflammatory reactions in diseases of and trauma to the central nervous system (CNS) [1]. TNF and its receptors, TNFR1 and TNFR2, are increased rapidly after spinal cord injury (SCI) [2,3,4,5,6,7,8] and high TNF concentrations have been linked to more severe SCI and worse functional outcomes [9,10]. TNF is synthesized by multifarious cells after SCI, including neurons, micro- and astroglia, oligodendrocytes, and infiltrating monocytes/macrophages [3,6,8,11]. TNF is involved in secondary tissue damage in the sub-acute phases but also in subsequent recovery in the chronic phases by affecting axonal and neuronal survival [1,4,11].

TNF is synthesized as a monomeric protein and becomes biologically active when it is incorporated into the membrane as a homotrimeric protein (tmTNF). tmTNF can be cleaved by TNF alpha converting enzyme (TACE/ADAM17) into a biologically active soluble trimer protein (solTNF) [12]. Due to differences in membrane-proximal extracellular stalk regions [13], TNFR1 and TNFR2 have different binding affinities for TNF, and diverse cytoplasmic tail structures result in distinct downstream signaling pathways [14]. TNFR1 is expressed by most cell types and has a higher affinity for solTNF than for tmTNF, leading to classical nuclear factor-kappa B (NF-κB)-mediated transcription of inflammatory genes, but it also contains a death domain that can either lead to caspase-mediated apoptosis or caspase-independent necroptosis [14,15,16]. TNFR2 is preferentially found on immune cells, glial cells, and endothelial cells, is primarily activated upon binding of tmTNF, and is linked to immune modulation and cell proliferation and survival [14,17].

Conflicting outcomes of manipulating TNF signaling are described in SCI, with some studies suggesting a neuroprotective outcome and other studies a detrimental outcome [18,19,20,21,22]. This dichotomy is likely due to differences in temporal and cellular expression of TNF after SCI [23,24] and, more importantly, in the levels of solTNF versus tmTNF. We and others have shown that interfering with solTNF-TNFR1 signaling is therapeutic in experimental SCI [25,26,27], while other studies have demonstrated that continuous infusion of a TNFR2 agonist improves recovery [28]. While some beneficial effects have been demonstrated using non-selective TNF therapeutics in preclinical SCI models [19,21,29,30,31,32], these molecules are also known to cause adverse effects in patients [33,34]. As inhibition of tmTNF signaling is thought to be the cause of these side effects, more selective TNF inhibitors have been developed. One of the approaches is selectively targeting solTNF-TNFR1 signaling, which is believed to be responsible for the detrimental effect of TNF signaling [32]. We previously demonstrated that selective inhibition of solTNF using the dominant-negative inhibitor XPro1595 reduced lesion volume and improved functional outcomes in mice subjected to SCI, probably by targeting microglial responses [26]. As we did not previously extensively study temporal neuroinflammatory responses, in the present work, we extended our observations by investigating the potential of selective solTNF inhibition on inflammatory and protective processes after SCI.

## 2. Materials and Methods

### 2.1. Mice

Female C57BL/6J mice (7–8 weeks old) were purchased from Taconic A/S (Ry, Denmark) and allowed to acclimatize for at least one week before surgery. All mice were group-housed under conventional conditions on a 12 h light/dark cycle with ad libitum access to food and water.

### 2.2. Contusive SCI

Mice were anesthetized with a cocktail (intraperitoneal (i.p.) injection) of ketamine (100 mg/kg, VEDCO Inc., Saint Joseph, MO, USA) and xylazine (10 mg/kg, VEDCO Inc.), and a laminectomy was performed at the eighth to the tenth thoracic vertebra (T8–T10), based on anatomical landmarks [35]. Mice received a moderate contusion injury (75 Kdyn) at the ninth thoracic (T9) level using the Infinite Horizon Device (Precision Systems and Instrumentation, Brimstone, LN, USA), as previously described [26]. Sham-operated mice only received the laminectomy. Mice were housed in individual cages in a recovery room at approximately 25 °C with a 12 h light/dark cycle until their wounds were healed. Thereafter, mice were group-caged and observed twice daily for activity level, respiratory rate, and general physical conditions. Mice surviving more than 24 h after surgery were weighed 1, 3, and 7 days after surgery and thereafter weekly. Bladders were manually emptied twice a day for the duration of the experiments. One mouse died during surgery.

### 2.3. Drug Treatment

Immediately after surgery, mice were implanted with micro-osmotic pumps (Alzet model 1003D, Durect Corporation, Cupertino, CA, USA), which, for a period of three days, continuously epidurally delivered XPro1595 (2.5 ng/mL/1 μL/h, INmune Bio Inc., Monrovia, CA, USA) or saline (0.9% physiological saline/1 μL/h), as previously described [26]. The pumps were placed in such a way that the delivering end of the catheter (Micro-Renathane Tubing MRE-040, AgnTho’s AB, Lindingö, Sweden) was on top of the injured spinal cord. The catheter was sutured to the musculature, and the suture and placement of it were secured using Vetbond (3M Animal Care Products, St. Paul, MN, USA). The pump was installed in a subcutaneous pocket on the lateral back of the mouse and kept in situ throughout the entire experiment.

Mice were allowed to survive 0 h (naïve, n = 10), 1 h (n = 5/treatment group), 3 h (n = 3), 1 day (n = 15/treatment group), 3 days (n = 15/treatment group), 7 days (n = 15/treatment group), 14 days (n = 10/treatment group), 21 days (n = 13/treatment group), 24 days (n = 3/treatment group), 28 days (n = 12/treatment group), or 35-days (n = 12/treatment group) after SCI. Saline-treated mice with 3 h survival are also part of another study on inflammatory markers in SCI [6].

### 2.4. Basso Mouse Scale (BMS)

Functional recovery of hindlimb function was assessed using the Basso Mouse Scale (BMS) and BMS subscore systems, as previously described [36]. Mice were evaluated over a 4 min period at 1, 3, and 7 days after SCI and weekly thereafter. Before surgery, mice were handled and pre-trained in the open field to prevent fear and/or stress behaviors that could bias the locomotor assessment. SCI-operated mice with a BMS score above 1 on day one after SCI were excluded from the study. One mouse was euthanized on day one after SCI due to a BMS score above 1.

### 2.5. Biotinylated Dextran Amine (BDA) Tracing

On day 14 after SCI, a group of mice were re-anesthetized with ketamine and xylazine and placed in a stereotaxic frame, a midline incision was made to reveal the bregma. Four small holes were drilled in the skull with a microdrill, and 0.5 µL 10% biotinylated dextran amine (BDA, cat. no. N-7167, Molecular Probes, Roskilde, Denmark) was injected over 2–4 min using a blunt 35 G needle (World Precision Instruments, cat. no. NF35BL-2) in both cerebral cortices using the following coordinates: 0.5 mm and −0.5 mm lateral; 1.0 mm and 1.5 mm posterior; with a depth of 0.5 mm from the cortical surface (n = 3/group). After injection, the skin overlying the skull was sutured, postoperative care was applied, and mice were replaced in their cages for an additional 10 days.

### 2.6. Tissue Processing

Real-time reverse transcriptase quantitative polymerase chain reaction (real-time RT-qPCR): One centimeter of the spinal cord was collected from mice at 0 (naïve), 1, 3, 7, 14, 21, and 35 days after SCI (n = 5/treatment group), snap-frozen on dry ice, and stored at −80 °C until further processing, as previously described [6].

*In situ hybridization:* One centimeter of spinal cord tissue centered on the lesion from mice that had survived for 3 h after SCI (n = 3) was dissected out and processed for in situ hybridization, as previously described [6].

*Protein analysis:* One centimeter of spinal cord tissue centered on T9 was collected from mice allowed to survive 1 h, 1, 3, 7, and 35 days (n = 4–5/treatment group) after SCI and naïve controls, as previously described [37]. Tissue was quickly snap-frozen on dry ice and stored at −80 °C until further processing.

*Immunohistochemistry and immunofluorescence staining:* Spinal cords from mice surviving 21, 28, or 35 days (n = 2–3/treatment group) after SCI were processed for immunohistochemistry and immunofluorescence staining, as previously described [6].

*Flow cytometry:* Tissue segments from mice surviving 1, 3, 7, 14, and 21 days after SCI (n = 5/treatment group) that contained the lesion area (1 cm centered on the lesion) and the peri-lesion area (0.5 cm distal and 0.5 cm proximal to the lesion were pooled to represent peri-lesion area) were processed for flow cytometry as previously described [6].

### 2.7. Phagocytosis Assay

To test the effect of selective solTNF inhibition on microglial phagocytic activity, primary microglia derived from postnatal day 0–2 (P0–P2) old C57BL/6J mouse brains (n = 21) were plated, activated, and incubated with fluorescent beads essentially as previously described [38]. First, the meninges were removed, and brains were collected in ice-cold Hank’s Balanced Salt Solution (HBSS) without Ca^2+^ and Mg^2+^ (ThermoFisher Scientific, cat. no. 55021C, Merck KGaA, Roskilde, Denmark) under sterile conditions. Cortices were processed using Neural Tissue Dissociation Kit (P) (Miltenyi Biotech, cat. no. 130-092-628, Lund, Sweden) according to the manufacturer’s protocol. Cell suspensions were filtered through 70 μm cell strainers (AH Diagnostics, cat. no. 352350, Aarhus, Denmark) and washed with HBSS with Ca^2+^ and Mg^2+^ (Gibco, cat. no. 24020-091). Cell suspensions were centrifuged, and the pellet, containing microglia, was used for magnetic-activated cell sorting (MACS sorting) using magnetic CD11b^+^ beads and LS columns (Miltenyi Biotech, cat. no. 130-042-401). Microglia were then cultured on coverslips in culture medium A (76% Dulbecco’s modified Eagle’s medium (DMEM) (ThermoFisher Scientific, cat. no. 11514435), 20% heat-inactivated FBS (Gibco, cat. no. 10108-165), 1% non-essential amino acids (NEAA) (ThermoFisher Scientific, cat. no. 11350912), 1% penicillin streptomycin (Pen Strep, Gibco, cat. no. 15140-106), 1% GlutaMAX (ThermoFisher Scientific, Gibco, cat. no. 35030038) and 1% sodium pyruvate (Gibco, cat. no. 11360039)) on poly-L-lysine-coated (Sigma-Aldrich, cat. no. P1274, Søborg, Denmark) plates in a humified CO_2_ incubator at 37 °C. Two days later, the medium was changed to 10% FBS-DMEM complete medium (10% heat-inactivated FBS, 86% DMEM, 1% Pen Strep, 1% Sodium Pyruvate, 1% GlutaMAX, and 1% NEAA). Three days after the media change, microglia were activated by adding 100 ng/mL lipopolysaccharide (LPS) (*E. Coli* O111:B4, Sigma-Aldrich, cat. no, L2630) and treated with either 100 or 200 ng/mL Xpro1595, or 100 or 200 ng/mL etanercept (ENT; Enbrel, Pfizer ApS, Ballerup, Denmark). LPS-stimulated cells were kept as controls. Furthermore, in a separate study, unstimulated cells and unstimulated cells treated with either Xpro1595 or etanercept were included as additional controls. Microglia were then incubated at 37 °C for 24 h. The following day, FluoSpheresTM Carboxylate-Modified Microspheres (1.0 mm, 505/515, yellow-green fluorescent, ThermoFisher Scientific, cat. no. F8813) were added to the cultured cells. After 2 h of incubation, phagocytosis was terminated by adding ice-cold DMEM for 2–5 min. Cells were fixed with 4% paraformaldehyde (PFA), blocked in 5% normal goat serum (VWR, cat. no. S2000-500) in Tris-buffered saline (TBS), and incubated overnight in anti-Iba1 (ionized calcium-binding adaptor molecule 1) antibody (1:500, Wako, cat. no. 019-19741, Osaka, Japan). The next day, cells were stained using Alexa 594-conjugated anti-rabbit IgG (1:750, Invitrogen, cat. no. 1A21207, Roskilde, Denmark) and 4′,6-diamidino-2-phenylindol (DAPI, 1:1000, ThermoFisher Scientific, cat. no. Sigma-Aldrich 09542), and mounted using ProLong Diamond (Invitrogen, cat. no. P36965). In total, 12–15 images were randomly taken from each coverslip at 20× magnification using an Olympus FluoView 1000 confocal microscope to ensure that it was possible to analyze 8 images per coverslip.

As a measure of phagocytic activity, the number of engulfed beads was counted, and morphological changes were estimated as previously described [38,39]. Three independent experiments were performed, and the results are presented as technical replicates.

### 2.8. Gene Analysis

Total RNA was isolated from spinal cord tissues using TRIzol reagent (Invitrogen, cat. no. 15596018) according to the manufacturer’s protocol and as previously described [6]. The purity and concentration of the RNA were analyzed using a ThermoFisher Scientific NanoDrop One Spectrophotometer.

Synthesis of cDNA was performed using the High-Capacity cDNA Reverse Transcription kit from Applied Biosystems (ThermoFisher, cat. no. 4368814). Equal amounts of RNA sample (2 µg RNA) and 2× RT Master mix were mixed into a 96-well plate, and after a short centrifugation cDNA was synthesized with the MJ Research PTC-225 Gradient Thermal Cycler from Marshall Scientific, using the following cycle conditions: 25 °C for 10 min, 37 °C for 120 min, 85 °C for 5 min, and then cooled down to 4 °C. Samples were diluted to 50 ng/µL and stored at −20 °C until further processing.

Real-time RT-qPCR was performed for each sample using primers for *Tnf*, *Tnfrsf1a*, *Tnfrsf1b*, *Il1b*, *Il6*, *Il10,* chemokine (C-X-C motif) ligand 1 (*Cxcl1*), integrin subunit alpha M (*Itgam*), c-x3-c motif chemokine receptor 1 (*Cx3cr1*), triggering receptor expressed on myeloid cells 2 (*Trem2*), purinergic receptor P2Y (*P2ry12*), arginase 1 (*Arg1*), chemokine (C-C motif) ligand 7 (*Ccl7*), cluster of differentiation (*Cd*)*-68*, *Cd8*, *Cd4*, *Cd3*, forkhead box P3 (*Foxp3*), glutamate ionotropic receptor AMPA type subunit 2 (*Gria2*), and hypoxanthine-guanin-phosphoribosyl-transferase 1 (*Hprt1*). Primer sequences are listed in Table 1. Primers were designed with NCBI’s nucleotide database and primer designing tool and aimed to target exon-exon junctions whenever possible. After a check for self-complementarity with OligoCalc [40], primers were purchased from TAG Copenhagen (Copenhagen, Denmark). All samples were analyzed using SYBRGreen (ThermoFisher Scientific, cat. no. KO223) and performed in a total volume of 12.5 µL containing 1× Maxima SYBRGreen, 50 ng of template cDNA, and forward and reverse primer (primer concentration: 600 nM, except for *Ccl7*: 900 nM). The amplification was carried out using a CFX Connect RealTime PCR Detection System from Bio-Rad under the following conditions: 95 °C for 10 min, followed by the optimal annealing temperature (T_a_) for 30 s and raised to 72 °C for 30 s, for the appropriate number of cycles (see T_a_ and the number of cycles for each gene in Table 1). The product was analyzed using melting curve analysis to ensure specificity. All samples were analyzed in triplicate, and the standard curve was prepared from a mixture of all tested samples with a 4-fold serial dilution. The standard curve samples were run in each assay together with the experiment samples, a calibrator (a mixture of all tested samples in 6 wells), and a no-template control. The experimental samples were randomly distributed over assays, primer efficiencies were accepted within the range of 100 ± 5%, and the relative transcript levels were calculated by the *Pfaffl* method [41] and normalized to the reference gene *Hprt1*. The temporal cycle threshold (Ct) values for *Hprt1* were found to be stable around 19 cycles and were comparable between treatment groups.

### 2.9. Protein Purification

Samples were thawed on ice and lysed in Complete Mesoscale Lysis Buffer (pH 7.5; 150 mM sodium chloride (Sigma-Aldrich, cat. no. 1064041000), 20 mM Tris, 1 mM Ethylene Diamine Tetra Acetate (EDTA, Sigma-Aldrich, cat. no. E9884), 1 mM ethylene glycol tetraacetic acid (EGTA, Sigma-Aldrich cat. no. E43781% Triton-X-100 (Merck, cat. no. X100), a cocktail of phosphatase and proteinase inhibitors (Sigma-Aldrich, cat. no. P5726 and Sigma-Aldrich, cat. no. P0044), and Complete, Mini, EDTA-free tablets (Roche, cat. no. 11836170001)), and tip-sonicated. After being shaken on ice at 4 °C for 30 min, samples were centrifuged at 14,000× *g* at 4 °C for 20 min, and the supernatants were stored at −80 °C until further analysis. The protein concentration was determined using the Pierce Bicinchoninic acid (BCA) Protein Assay Kit (ThermoFischer Scientific, cat. no. 23235) according to the manufacturer’s protocol.

### 2.10. Electrochemiluminescence Analysis

XPro1595, TNF, IL-1β, IL-6, IL-10, CXCL1, interferon-gamma (IFNγ), TNFR1, and TNFR2 protein levels were investigated in spinal cord protein lysates using custom-made MSD Mouse Pro-inflammatory V-PLEX and Ultra PLEX kits (Mesoscale Discovery Rockville, MD (1 and 24 h and 3 and 7 days after SCI and naïve controls: cat. no. K152BIC; 35 days after SCI and naïve controls: cat no. K152AOH-2)) and Ultra-sensitive TNFRI (Mesoscale Discovery (1 and 24 h and 3 and 7 days after SCI and naïve controls: cat. no. K152BIC; 35 days after SCI and naïve controls: cat. no. K152AOH-2)), and TNFRII (Mesoscale Discovery (1 and 24 h and 3 and 7 days after SCI and naïve controls: cat. no. K152BIC; 35 days after SCI and naïve controls: cat. no. K152AOH-2)) kits, as previously described [23]. Analysis of tissue derived from mice with survival times 1 and 24 h and 3 and 7 days after SCI was performed separately from mice with 35 days survival after SCI, and therefore analyzed as two separate experiments. Samples were diluted in Diluent 41 according to the manufacturer’s instructions, run in duplex on a SECTOR Imager 6000 Plate Reader (Mesoscale), and analyzed using MSD Discovery Workbench software. Samples with coefficient of variation (CV) values > 25% in individual analyses were excluded. The lower limit of detection (LLOD) was a calculated concentration based on a signal of 2.5 standard deviations (SD) above the blank (zero) calibrator. For protein levels below LLOD, a value of 0.5 LLOD was used for statistical analysis. LLOD values for samples with naïve conditions, 1 and 24 h and 3 and 7 days survival after SCI: XPro1595 = 18.40 pg/mL, IL-1β = 0.17–0.22 pg/mL, IL-10 = 0.80–1.34 pg/mL, CXCL1 = 0.14–0.23 pg/mL, TNF = 0.22–0.77 pg/mL, IL-6 = 1.70–4.04 pg/mL, TNFR1 = 0.52–0.61 pg/mL, and TNFR2 = 15.00–35.9 pg/mL. LLOD values for samples with 35 days survival after SCI and naïve: IL-1β = 0.11 pg/mL, IL-10 = 0.27 pg/mL, CXCL1 = 0.12 pg/mL, TNF = 0.22 pg/mL, IL-6 = 2.04 pg/mL, IFNγ = 0.06 pg/mL, TNFR1 = 0.18 pg/mL, and TNFR2 = 0.62 pg/mL.

### 2.11. Automated Western Immunoblotting

The Jess Simple Western system (ProteinSimple, San Jose, CA, USA), an automated capillary-based size separation and nano-immunoassay system, was used to quantify postsynaptic density protein 95 (PSD95) and myelin basic protein (MBP) levels in saline- and XPro1595-treated mice 35 days after SCI (n = 5/group), as well as in naïve mice (n = 6). To quantify PSD95 and MBP levels, 0.4 mg/mL protein was loaded on 12–230 kDa Jess separation modules (Biotechne, cat. no. SM-W004, Dublin, Ireland) according to the manufacturer’s instructions. Primary antibodies used were rabbit anti-PSD95 (1:100, Abcam, cat. no. ab18258, Cambridge, United Kingdom), rat anti-MBP (1:50, Merck, cat. no MAB386, Søborg, Denmark), and α-actin (1:100, Merck, cat. no. MAB1501, Søborg, Denmark). Secondary antibodies used were HRP-conjugated anti-rabbit IgG (1:20, Bio-Techne, cat. no 043-426, Dublin, Ireland), HRP-conjugated anti-rat IgG (1:50, Bio-Techne, cat. no HAF005), and HRP-conjugated anti-mouse IgG (ready to use, ProteinSimple, Biotechne, cat. no. 042-205, Dublin, Ireland). Data were analyzed using Compass for Simple Western (v5.0.1). Signal-to-noise values above 10 were accepted for analysis, and the protein expression was measured as peak area. PSD95 and MBP levels were normalized using α-actin, and data are presented as percentages relative to naïve mice.

### 2.12. In Situ Hybridization

In situ hybridization was performed on EtOH-fixed spinal cord sections and the hybridization signal was developed with an alkaline phosphatase (AP) buffer containing 5-bromo-4-chloro-3-indolyl phosphate (Sigma-Aldrich, cat. no. B8503) and nitroblue tetrazolium (Sigma-Aldrich, cat. no. N6876), as previously described [6,42]. A mixture of two AP-labeled oligo DNA probes (3 pmol/mL) was applied to tissue sections from mice with 3 h survival after SCI (n = 3). Probes were purchased from DNA Technology (Copenhagen, Denmark); *Tnf* probes: 5′ CGTAGTCGGGGCAGCCTTGTCCCTTGAA 3′ (GC content 60.7%, Tm 67.8 °C) and 5′ CTTGACGGCAGAGAGGAGGTTGACTTTC 3′ (GC content 53.6%, Tm 62.3 °C), and glyceraldehyde 3-phosphate dehydrogenase (*Gapdh*) probe: 5′ CCTGCTTCACCACCTTCTTGATGTCA 3′ (GC content 50%, Tm = 60.2 °C). Abolishment of the hybridization was confirmed by hybridizing RNase A-digested sections and the absence of signal was confirmed by hybridizing sections with 100-fold excess of the unlabeled probe mixture or by incubating with buffer only [6].

### 2.13. Immunostaining and Image Analysis

For immunostaining, sections were rinsed with TBS and TBS with 0.5% Triton X-100, followed by blocking with 10% FBS in TBS with 0.5% Triton X-100 for 30 min and incubated with the following antibodies in different combinations overnight at 4 °C: chicken anti-MAP2 (Microtubule-associated protein 2, 1:100, Abcam, cat. no. ab5392), rabbit anti-TNFR1 (1:50, clone H-271, Santa Cruz, cat. no. sc-7895, Århus, Denmark), rabbit anti-TNFR2 (1:200, Sigma-Aldrich, HPA004796), rat anti-MBP (1:200, 1:50, Merck, cat. no MAB386), rabbit anti-Iba1 (1:500, Wako, cat. no. 019-19741, Osaka, Japan), rat anti-CD68 (1:400, Bio-Rad, cat. no. MCA1957, Copenhagen, Denmark), and rat anti-Gal3 (Galectin-3, 1:300, clone M38, Hakon Leffler’s Lab [43]). On the following day, sections were rinsed in TBS and incubated with the appropriate secondary antibodies for 2 h at RT: donkey anti-rabbit (1:200, Alexa-594, Invitrogen, cat. no. A21207), chicken anti-rabbit (1:200, Alexa-488, Invitrogen, cat. no. A21441), goat anti-rat (1:200, Alexa-594, Invitrogen, cat. no. A11007), goat anti-rat (1:200, Alexa-488, Invitrogen, cat. no. A11006), or goat anti-chicken (1:200, Alexa-488, Invitrogen, cat. no. A11039). Sections stained using anti-GFAP-Cy3 (Glial fibrillary acidic protein, 1:500, clone G-A-5, cat. no. C9205, Sigma-Aldrich) or anti-GFAP-488 (1:400, clone 131-17719, cat. no. A21294, Invitrogen) were only incubated 1 h. Finally, sections were rinsed with TBS, counterstained with DAPI (1:1000, Sigma-Aldrich, cat. no. 09542,), and mounted with Aquatex (Merck, cat. no. HC718601, Søborg, Denmark).

The quantitative analysis of Iba1 and CD68 immunofluorescence was carried out on images taken within 0–500 µm from the lesion border in 3 sections per mouse (n = 3/treatment group). Briefly, pictures of Iba1 and CD68 immunostained sections were acquired at 10X magnification, converted to binary grayscale to better visualize microglial cell morphology, and analyzed using the particle analysis tool of the Fiji-ImageJ software as previously described [44]. The density of Iba1^+^ and Iba1^+^/CD68^+^ cells was calculated as the number of particles/area analyzed (mm^2^) and expressed as mean ± standard error of the mean (SEM). The area fraction covered by Iba1 and CD68 staining, and the average size of Iba1^+^ cells, were automatically determined and are expressed as fold-over values obtained in saline control mice set to 100. All images were acquired using an Olympus BX53 fluorescence microscope fitted with an Olympus DP73 camera, and images were merged and adjusted in their brightness/contrast levels in Photoshop.

BDA-labeled cells and axons were detected by application of horse-radish-peroxidase (HRP)-Streptavidin. Briefly, the sections were rinsed with TBS and blocked for endogenous peroxidase with methanol containing H_2_O_2_. Next, sections were rinsed in TBS, then in TBS with 0.5% Triton X-100, pre-incubated with 10% FBS in TBS with 0.5% Triton X-100, and incubated with HRP-Streptavidin (1:200, Amersham, cat. no. RPN1231, Søborg, Denmark). Sections were developed using diaminobenzidine (DAB, Santa Cruz, cat. no. sc-216567A, Heidelberg, Germany) dissolved in TBS containing H_2_O_2_ and mounted with Aquatex. Tissue sections from a mouse with SCI, with no injected BDA tracer, were used as negative controls. Images were scanned with a NanoZoomer slide scanner from Hamamatsu. The total number of BDA^+^ cells per mm^2^ was calculated in the whole spinal cord from each animal by counting all BDA^+^ cells 2 mm rostral and caudal to the center of the lesion, and area was estimated using the NDP.view2 software (n = 3/treatment group) [45].

### 2.14. Flow Cytometry

For flow cytometry, spinal cord tissue from each mouse was processed individually using the following panel: PerCP-Cy5.5 rat anti-CD45 (1:100, clone 30-F11, BD Biosciences, cat. no 561869, Lyngby, Denmark), PE rat anti-CD11b (1:200, clone M1/70, BD Biosciences, cat. no 564454), PE-Cy7 rat anti-Ly-6C (1:200, clone AL-21, BD Biosciences, cat. no. 560593), BV421 rat anti-LY-6G (1:200, clone 1A8, BD Biosciences, cat. no. 562737), APC hamster anti-CD3 (1:100, clone 145-2C11, BD Bioscience, cat. no. 553066) or the corresponding isotype controls PerCP-Cy5.5 rat IgG2b,κ (clone A95-1, BD Biosciences, cat. no. 550764), PE rat IgG2_b_,κ (clone A95-1, BD Biosciences, cat. no. 564421), PE-Cy7 rat IgG2c,κ (clone, R35-95, BD Biosciences, cat. no. 560572), BV421 rat IgG2a,κ (clone, R35-95, BD Biosciences, cat. no. 562602), and APC Armenian hamster IgG1,κ (clone HTK888, Biolegend, Nordic Biosite, cat. no. 553975, Täby, Sweden) to identify the following cell populations: microglia (CD11b^+^CD45^dim^), macrophages (CD11b^+^CD45^high^Ly6C^high^Ly6G^−^), granulocytes (CD11b^+^CD45^high^Ly6C^+^Ly6G^+^), and T cells (CD45^+^CD3^+^), as previously described [46]. Samples collected on day 14 were run using the same protocol but without the CD3 marker and using different fluorophore combinations and therefore are presented separately.

After homogenization of the tissue, cells were resuspended in PBS with 10% FBS and the myelin was removed using Myelin Removal Beads II (Miltenyi Biotec, cat. no. 130-096-731) with LS column (Miltenyi Biotec, cat. no. 130-042-401) placed in the magnetic field of the MACS separator (Miltenyi Biotec). Red blood cells were lysed, and the cell suspensions were stained for live/dead cells using Fixable Viability Dye eFlouro 506 (eBioscience, cat. no. 65-0866-18, San Diego, CA, USA), washed, and fixed with Cytofix/Cytoperm (BD Biosciences, cat. no. 554714) as previously described [46]. Next, cell suspensions were centrifuged at 300× *g* for 10 min at 4 °C, the supernatants removed, and the pellet resuspended in flow cytometry staining buffer (FACS-buffer). Thereafter, cells were centrifuged at 600× *g* for 10 min at 4 °C and blocked to prevent non-specific staining (anti-CD16/32, FcR block, BD Biosciences, cat. no. 553141) for 30 min at 4 °C. After washing, the samples were resuspended in FACS-buffer containing antibodies for the desired surface markers or their corresponding isotype controls. Samples were run on a FACSverse flow cytometer, and approximately 10^5^ events were acquired per sample using forward scatter (FSC) and side scatter (SSC). The analysis was performed using the FACSuite software as previously described [47], and the mean fluorescence intensity (MFI) was calculated as the geometric mean of each population in the CD45 and CD11b positive gates. Positive staining was determined based on the respective isotype controls and respective fluorescent minus one (FMO) control. Total cell numbers were calculated by adjusting for the fraction of the sample collected (determined by measuring the sample volume before and after acquisition), as previously described.

### 2.15. Statistical Analysis

Generally, comparisons were performed using repeated measures (RM) or regular two-way analysis of variance (ANOVA) followed by Sidak’s *post hoc* analysis, ordinary one-way ANOVA followed by Dunnett’s *post hoc* analysis, or by Student’s *t*-test. Outliers were identified using ROUT with a False Discovery Rate (FDR) of 1%. BMS was analyzed using two-way ANOVA and Fisher’s Least Significant Difference (LSD) test using a 1% FDR. Correlation analysis was performed using Spearman correlation analysis. All analyses were performed using Prism 4.0b software for Macintosh (GraphPad Software, San Diego, CA, USA). Statistical significance was established by *p* < 0.05. Data are presented as mean ± SEM.

## 3. Results

### 3.1. Inhibition of solTNF Improves Functional Outcome after SCI

As we previously demonstrated that topical XPro1595 treatment for three consecutive days significantly reduced lesion volumes and improved functional outcomes after SCI in mice [26], we initially wanted to confirm the therapeutic efficacy of XPro1595 in our model. We found BMS scores to be comparable between saline- and XPro1595-treated mice on day 1 and day 3, with XPro1595-treated mice displaying improved hindlimb function from day 7 onwards, compared to saline-treated mice (Figure 1a). BMS subscores improved significantly from day 14 after SCI in XPro1595- compared to saline-treated mice (Figure 1b). We observed no differences between treatment groups in temporal urine content or temporal changes in total body weight (Appendix A). Using electrochemiluminescence, we confirmed that XPro1595 successfully reached the injured spinal cord tissue in therapeutically relevant doses, with XPro1595 levels peaking 3 days after SCI, whereafter levels declined (Figure 1c).

At 21, 28, and 35 days after SCI, intense GFAP immunoreactivity was detected around the lesion and in the surrounding white and grey matter in both treatment groups, indicating the formation of a glial scar (Figure 1d). The lesion area was filled with Iba1^+^ myeloid cells packed at a high density in both treatment groups (Figure 1d), surrounded by ramified Iba1^+^ microglial cells in the peri-lesion areas (insert in Figure 1d).

### 3.2. Inhibition of solTNF Does Not Affect the Spatio-Temporal Expression of TNF after SCI

As *Tnf* is known to act in an autocrine manner both at the transcriptional and protein level [48], we wanted to assess whether XPro1595 treatment influenced the temporal expression levels of *Tnf* mRNA and TNF protein after SCI. We performed real-time RT-qPCR and electrochemiluminescence analyses. *Tnf* mRNA (Figure 2a) and TNF protein (Figure 2b,c) levels increased significantly in the lesioned area after SCI, but no differences were observed between treatment groups. TNF levels in sham mice remained low, with no differences between treatment groups (Appendix A). Then we also wanted to confirm the cellular and spatial localization of TNF in our model [6]. Using in situ hybridization, we found that by 3 h after SCI, *Tnf* mRNA^+^ cells were located primarily in the white matter of the posterior funiculi (arrows top image in Figure 2d, shown for saline-treated mice only), with also a few *Tnf* mRNA^+^ cells, presumably neutrophils or macrophages, located inside blood vessels (Figure 2d, bottom image). Most *Tnf* mRNA^+^ cells had a glial-like morphology (arrows in Figure 2d,e); these were presumably microglia as they did not co-express the astrocytic marker GFAP (arrowheads in Figure 2e). By 35 days after SCI, TNF co-localized to CD11b^+^ cells located within the lesion core, just as TNF expression was significantly increased on CD11b^−^ cells surrounding the lesion (Figure 2f).

### 3.3. Inhibition of solTNF Does Not Affect the Spatio-Temporal Expression of TNF Receptors after SCI

We previously showed that the protective effects of inhibiting pro-inflammatory solTNF function were associated with the upregulation of neuroprotective TNFR2 expression in the spinal cord [26]. Thus, we assessed the effect of inhibition of solTNF signaling on TNF receptor gene expression and protein levels, by performing real-time RT-qPCR and electrochemiluminescence analyses in saline- and XPro1595-treated mice after SCI (Figure 3).

*Tnfrsf1a* mRNA levels increased significantly from 1 day after SCI, compared to naïve conditions (Figure 3a). In XPro1595-treated mice, *Tnfrsf1a* mRNA reached peak levels at day 3 after SCI, while in saline-treated mice *Tnfrsf1a* mRNA reached peak levels at day 7, suggesting accelerated *Tnfrsf1a* expression following solTNF inhibition. TNFR1 levels increased significantly by day 1 after SCI, with the highest levels detected 3 days after SCI (Figure 3b,c). No significant differences were observed between treatment groups. In sham mice, TNFR1 levels increased transiently at 1 day, with no differences between treatment groups (Appendix A). Similar to previous findings [6], immunofluorescent double labeling for TNFR1 and the neuronal marker MAP2 showed that TNFR1 co-localized to MAP2^+^ neurons and their proximal dendrites near the border of the lesion at days 21, 28, and 35 after SCI in both saline- and XPro1595-treated mice (Figure 3d,e). Furthermore, TNFR1 was expressed by MAP2^−^ cells surrounding the lesion core (Figure 3d,e).

*Tnfrsf1b* mRNA levels also increased significantly after SCI, with the highest expression levels detected on days 3–7 (Figure 3f). No differences were observed between treatment groups. TNFR2 levels increased significantly from day 1 after SCI, with no significant differences observed between treatment groups (Figure 3g,h). In sham mice, TNFR2 levels transiently increased at 1 day with no significant differences between treatment groups (Appendix A). Similar to previous findings [6], immunofluorescent double labeling for TNFR2 and astroglial GFAP showed high co-localization, especially near the lesion, at 21, 28, and 35 days after SCI in both treatment groups, suggesting high TNFR2 expression in the peri-lesion scar composed by glial and fibrotic cells (Figure 3i,j). Furthermore, TNFR2 was expressed by GFAP^−^ cells, presumably macrophages, located within the lesion core (Figure 3i).

### 3.4. Inhibition of solTNF Alters the Inflammatory Environment after SCI

To further examine the effect of solTNF inhibition on the level of important inflammatory cytokines after SCI, we measured mRNA and protein levels of IL-1β, IL-6, IL-10, and CXCL1 using real-time RT-qPCR and electrochemiluminescence analyses (Figure 4a–l).

We found that *Il1b* mRNA levels increased after SCI, peaking on day 1 (Figure 4a). At this time point, we also observed the highest IL-1β levels (Figure 4b), although IL-1β levels in XPro1595-treated mice were significantly lower than in saline-treated mice (Figure 4b). Thirty-five days after SCI, IL-1β levels were still significantly elevated compared to naïve conditions, although at this time point IL-1β levels were significantly higher in XPro1595- compared to saline-treated mice (Figure 4c).

*Il6* (Figure 4d) and IL-6 (Figure 4e,f) levels increased significantly after SCI, peaking at 1 day, where also levels were significantly lower in XPro1595- compared to saline-treated mice (Figure 4d,e).

Though *Il10* mRNA levels increased significantly after SCI, no differences were observed between treatment groups (Figure 4g). In contrast, IL-10 levels decreased in the acute phase after SCI (Figure 4h) and increased in the more delayed phase post-SCI (Figure 4i). XPro1595 significantly increased IL-10 levels 3 days after SCI (Figure 4h), with no differences observed between treatment groups at any other time point (Figure 4h,i).

Even though *Cxcl1* (Figure 4j) and CXCL1 (Figure 4k,l) levels changed significantly over time after SCI, we observed no differences between treatment groups.

In sham mice, levels of IL-1β, IL-6, and CXCL1 increased transiently at day 1, but with no differences between treatment groups (Appendix A). IL-10 levels did not change in saline- or XPro1595-treated sham mice (Appendix A).

As TNF is implicated in the regulation of post-synaptic AMPA receptors and synaptic plasticity [49] and has been demonstrated to enhance the susceptibility of motor neurons to slow excitotoxic injury [50], we examined whether inhibition of solTNF affected transcription levels of the AMPA receptor subunit 2, *Gria2*, after SCI. Although *Gria2* mRNA levels decreased transiently 1 day after SCI, no differences were observed between treatment groups (Figure 4m).

CCL7 has been shown to contribute to TNF-dependent inflammation [51], just as TNF can stimulate neurons [52] and astrocytes to produce CCL7 [53], thereby modulating the inflammatory response. As XPro1595 has previously been shown to decrease CCL7 levels in the spinal cord of mice with experimental autoimmune encephalomyelitis (EAE) and these mice are protected from EAE [54], we studied the effect of selective solTNF inhibition on the temporal *Ccl7* expression. *Ccl7* mRNA levels increased transiently 1–3 days after SCI (Figure 4n), suggesting a role of this chemokine in the acute phase after SCI. However, no differences were observed between treatment groups (Figure 4n).

As IFNγ has been suggested to improve the outcome of traumatic SCI (reviewed in [55]), we measured IFNγ levels in the chronic phase after SCI (Figure 4o), when the functional outcome was significantly improved in XPro1595-treated compared to saline-treated mice (Figure 1a,b). We found a trend towards increased IFNγ levels in XPro1595-treated mice; however, this did not reach statistical significance (Figure 4o).

### 3.5. Inhibition of solTNF Decreases the Infiltration of Peripheral Immune Cells 14 Days after SCI

Using flow cytometry, we assessed the numbers and population distribution of microglia and infiltrating immune cells 1, 3, 7, 14, and 21 days after SCI (Figure 5 and Appendix A). Gating was performed to include live cells only, and gates were set to include CD45^dim^CD11b^+^ microglia and infiltrating CD45^high^CD11b^+^ leukocytes, which were further gated into Ly6C^+^Ly6G^−^ macrophages and Ly6C^+^Ly6G^+^ granulocytes (Figure 5a).

Although the number of microglia changed over time after SCI, we observed no differences between treatment groups (Figure 5b,c). The number of infiltrating leukocytes was highest 1 day after SCI (Figure 5d,e), with no differences between treatment groups (Figure 5e) except for day 14, where we observed a significant reduction in the number of infiltrating leukocytes in XPro1595- compared to saline-treated mice (Figure 5e). By differentiating the analysis further, we observed the same pattern for macrophages (Figure 5f,g) and granulocytes (Figure 5h,i). In the peri-lesion area, XPro1595 treatment transiently increased the number of microglia 14 days after SCI (Appendix A), whereas no differences in the number of infiltrating leukocytes were observed between treatment groups (Appendix A).

On days 1, 3, 7, and 21 after SCI, CD3 was included as a marker to estimate changes in T-cell numbers and population distribution (Figure 5j,k). As day 14 samples were run separately without the CD3 marker, we at this timepoint gated for CD45^high^CD11b^−^ lymphocytes (Figure 5l,m). Although the number of T cells increased after SCI, no differences were observed in the lesion area between treatment groups (Figure 5k–m). The same was true for the peri-lesion area (Appendix A).

As a measurement of cell activation, we estimated MFI for CD45 or CD11b on microglia, macrophages, and granulocytes in the lesion area (Appendix A) and peri-lesion area (Appendix A) after SCI. XPro1595 treatment decreased the MFI for CD11b on macrophages in the lesion area 14 days after SCI (Appendix A), and MFI for CD45 was decreased on microglia in the peri-lesion area 7 days after SCI (Appendix A). Changes in cell populations after SCI can be found in Appendix A.

### 3.6. Inhibition of solTNF Changes CD68 Microglial/Macrophage Activation State after SCI

To characterize the effect of solTNF inhibition on microglial/macrophage responses after SCI, we initially investigated gene expression levels of typical phenotypic microglia/macrophage markers using real-time RT-qPCR (Figure 6a–f). *Arg1*, known to be involved in wound repair [56], increased transiently 1–7 days after SCI, reaching peak levels by day 3, but no differences were observed between treatment groups (Figure 6a). *Itgam* also increased after SCI, reaching peak levels on day 3, and remained elevated throughout the study (Figure 6b). No differences were observed between treatment groups. *P2ry12* (Figure 6c), *Trem2* (Figure 6d), and *Cx3cr1* (Figure 6e) all significantly increased 3 days after SCI and remained elevated throughout the study, with no differences between treatment groups. *Cd68*, a general marker of activated phagocytic microglia/macrophages [57], also increased significantly after SCI, reaching peak levels at day 7, with a significant increase in saline- compared to XPro1595-treated mice (Figure 6f), suggesting altered activation state in CD68^+^ microglia/macrophages.

We previously demonstrated that XPRo1595 affects Iba1 protein levels in the spinal cord after SCI [26]. Thus, we used immunofluorescent staining to identify the expression pattern and co-localization of CD11b (Figure 6g), Gal3 (Figure 6h), and CD68 (Figure 6i,j) with Iba1^+^ microglia/macrophages in the delayed phase after SCI. The majority of Iba1^+^ microglia/macrophages in the lesion core and peri-lesion area co-labeled with CD11b, whereas in more distant areas, Iba1^+^ cells did not show a strong CD11b immunofluorescent signal (Figure 6g). Gal3 was mainly found on Iba1^+^ cells located within the lesion core but also in Iba1^−^ cells located in the peri-lesion area (Figure 6h). CD68 was found to preferentially co-localize to Iba1^+^ cells within the lesion core as well as to Iba1^+^ cells located in the peri-lesion area (Figure 6i), but to a lesser extent in XPro1595-treated mice (Figure 6j). Based on this observation, we thoroughly characterized the morphology of microglial-like Iba1^+^ and CD68^+^ cells in the peri-lesion area (Figure 6k–m). Conversion of high-magnification images to grayscale clearly indicated a difference in microglial activation states between saline- and XPro1595-treated mice 21 days post-SCI (Figure 6k). Although the Iba1^+^ cell density and area fraction were comparable between treatment groups, this difference was reflected in a significant decrease in Iba1^+^ cell average size in XPro1595-treated mice (Figure 6l). Finally, the density of Iba1/CD68^+^ cells was also comparable between treatment groups, but the CD68 area fraction was significantly decreased in XPro1595- compared to saline-treated mice. This indicates XPro1595 treatment after SCI reduces the activation state of CD68^+^/Iba1^+^ microglia located in the peri-lesion area.

### 3.7. Selective Inhibition of solTNF Increases Microglial Phagocytosis

To examine the role of selective solTNF inhibition on the phagocytic properties of microglia, we next examined how selective solTNF versus non-selective TNF inhibition in vitro affected microglial phagocytosis and cell morphology in unstimulated versus LPS-stimulated conditions (Figure 7).

Inhibition of solTNF using either 100 or 200 ng/mL XPro1595 in unstimulated primary microglia (Figure 7a) did not affect the number of engulfed beads per cell (Figure 7b), cell area (Figure 7d), perimeter length (Figure 7d), or membrane ruffling (Figure 7e), as compared to control microglia (Ctl). Following LPS stimulation, however, 200 ng/mL XPro195 (Figure 7f) significantly increased microglial phagocytosis (Figure 7g), without affecting cell area (Figure 7h), perimeter length (Figure 7i), or membrane ruffling (Figure 7j).

In comparison, although non-selective TNF inhibition using ETN (Figure 7k) did not affect microglial phagocytosis in un-stimulated microglia (Figure 7l), 200 ng/mL ETN significantly increased cell area (Figure 7m), suggesting increased activation state. Perimeter length (Figure 7n) and membrane ruffling (Figure 7o) were not affected by ETN in unstimulated conditions. Following LPS stimulation (Figure 7p), 200 ng/mL ETN significantly decreased microglial phagocytosis (Figure 7q), but in a concentration-dependent manner increased cell area (Figure 7r) and perimeter length (Figure 7s). Membrane ruffling was not affected by ETN treatment (Figure 7t). This suggests that non-selective inhibition of activated microglia leads to a decrease in microglial phagocytosis and an increase in microglial activation state.

### 3.8. Selective Inhibition of solTNF Does Not Affect the Density of Corticospinal Tract Fibers Post-SCI

To evaluate spared and/or regenerating corticospinal tract (CST) axons, we injected BDA into the motor cortex 14 days after SCI and investigated anterograde fiber tracing 24 days after SCI. As BDA becomes incorporated by neurons in the motor cortex and is anterogradely transported by axons through the descending corticospinal tract without interruptions, it provides the possibility to estimate the density of spared and/or regenerating axons in the injured spinal cord. By 24 days, we found BDA^+^ cells located within the lesion core of both saline- and XPro1595-treated mice (Figure 8a). As these cells are presumed to be phagocytic cells and their number in the lesion correlates to corticospinal tract density [45], we estimated the density of these cells. However, we found no differences in the density of BDA^+^ phagocytic-like cells between the two treatment groups (Figure 8b). We observed no BDA-labeled tract fibers caudal to and within the lesion of either group, possibly due to axonal dieback. BDA-labeled tract fibers were located rostral to the injury with no obvious differences between saline- and XPro1595-treated mice (Figure 8c). These findings were in line with observations of comparable post-synaptic PSD95 levels between saline- (61.00% ± 6.27%) and XPro1595-treated (70.81% ± 4.30%) mice 35 days after SCI (*p* = 0.23) (Figure 8d) and was reflected by a lack of correlation between PSD95 levels and functional outcome (BMS) (Figure 8e).

### 3.9. Selective Inhibition of solTNF Results in Preserved MBP Levels after SCI

Although XPro1595 treatment did not impact corticospinal tract density or improve neuronal plasticity, as measured by PSD95, we found a significant increase in MBP levels in XPro1595- (72.97% ± 10.22%) as compared to saline-treated (44.72% ± 6.52%) mice 35 days after SCI (*p* = 0.048) (Figure 8f). MBP levels also correlated significantly with improved functional outcomes (Figure 8g). Preservation of myelin integrity by XPro1595 was also confirmed by MBP immunostaining (Figure 8h).

### 3.10. Inhibition of solTNF Does Not Alter Gene Expression of Phenotypic T-Cell Markers after SCI

As selective inhibition of solTNF has been shown to reduce the infiltration of cytotoxic CD4^+^ T cells in experimental animal models with chronic neuroinflammation [58], we also investigated whether XPro1595 altered the expression of genes related to T cell populations (Figure 9). Although the expression of *Cd3* (Figure 9a), *Cd8* (Figure 9b), *Cd4* (Figure 9c), and *Foxp3* (Figure 9d) all increased after SCI, we observed no differences between treatment groups.

## 4. Discussion

In the present study, we extended our previous findings of a neuroprotective effect of selective solTNF inhibition on lesion size and functional outcome after SCI [26] to further study the effect on inflammatory and pro-regenerative processes after SCI. Initially, we verified our previous findings of improved functional outcomes in XPro1595-treated mice after SCI [26] and demonstrated that XPro1595 levels within the lesioned spinal cord reached concentrations predicted to be pharmacologically relevant [59,60]. Despite comparable TNF and TNF receptor levels, we found reduced levels of the pro-inflammatory cytokines IL-1β and IL-6 and increased levels of the anti-inflammatory cytokine IL-10 in the acute phase after SCI, suggesting a switch of the inflammatory response in the spinal cord of XPro1595-treated mice towards regenerative traits. Accordingly, we also observed a reduction in the number of infiltrating peripheral leukocytes 14 days after SCI and a smaller Iba1^+^ cell average size and CD68^+^ area fraction 21 days after SCI, indicative of a blunted reactivity of microglia/macrophages at chronic disease stages in XPro1595-treated mice. Moreover, our findings revealed an increased phagocytic capacity of XPro1595-treated LPS-stimulated primary microglia after acute exposure to XPro1595, which was completely abolished in etanercept-treated microglia. Finally, we observed increased IL-1β and MBP protein levels in the lesioned spinal cord 35 days after SCI in XPro1595-treated mice, of which the latter positively correlated with functional outcome. Our findings suggest that topical XPro1595 treatment after SCI transiently modifies the inflammatory response to favor a protective environment, resulting in increased myelin integrity, and improved functional outcome.

Although non-selective TNF inhibitors have demonstrated some efficacy in single cases of human SCI [61] and in preclinical SCI models (reviewed in [8]), the risk of adverse effects [33], including CNS demyelinating disease [34,62], exceeds the potential benefit of their use in neurological patients. This has increased the interest in more selective TNF therapeutics that target detrimental solTNF-TNFR1 signaling while preserving beneficial tmTNF-TNFR2 signaling [59,63]. In the present study, we verify previous findings by us and others demonstrating that inhibition of solTNF is beneficial in SCI [25,26]. Similar findings have been observed in other acute CNS injuries such as traumatic brain injury [64] and ischemic stroke [47,65]. In support of this, topical administration of EHD2-sc-TNF_R2_, a selective TNFR2 agonist, improved functional outcomes after SCI and increased neural cortical responses following hindlimb stimulation [28]. Additionally, neurons were protected from glutamate-mediated excitotoxicity through activation of PI3K in vitro [28]. In other studies, systemic administration of EHD2-sc-TNF_R2_ alleviated central and peripheral inflammation, reduced demyelination and neurodegeneration, improved motor symptoms, and promoted long-term recovery from neuropathic pain in an animal model of multiple sclerosis [66], while treatment with NewSTAR2, another selective TNFR2 agonist, ameliorated neuropathology and improved cognition in an Alzheimer’s disease mouse model [67]. Together, these studies highlight the importance of developing new selective TNF therapeutics that can be successfully used in neurological patients.

In line with previous studies (reviewed in [8]), TNF levels increased rapidly in the injured spinal cord. This increase was independent of selective solTNF inhibition, as a similar spatio-temporal expression of TNF was observed in XPro1595-treated mice. The same was true for TNFR1 and TNFR2, although we observed a transient decrease in *Tnfrsf1a* at 7 days after SCI in XPro1595-treated mice, compared to saline-treated mice. We previously demonstrated sustained TNFR2 levels in XPro1595-treated mice 7 days after SCI [26], but other studies have shown that inhibition of solTNF signaling does not affect TNF or TNFR2 levels [46,58,64,65], suggesting that selective inhibition of solTNF-TNFR1 signaling induces protective effects independent of the levels of TNF and its receptors in the lesioned spinal cord.

One of the mechanisms by which the acute production of solTNF is suggested to be cytotoxic is through synergistic interactions with glutamate [68,69]. The distribution of post-synaptic glutamate receptors has been shown to be regulated by TNF signaling [70], and TNF has been shown to trigger rapid membrane insertion of AMPA receptors and, in some cases, specific insertion of GluR2-lacking, Ca^2+^ permeable AMPA receptors into motor neurons, enhancing their susceptibility to slow excitotoxic injury [50]. We did not, however, find temporal changes in GluR2 AMPA receptor expression levels with XPro1595 treatment, as *Gria2* levels were comparable between XPro1595- and saline-treated mice. This is in line with a previous study demonstrating that despite findings of ameliorated synaptic alterations and Ca^2+^ dysregulation in aged rats following XPro1595 treatment, this change was independent of changes in GluR2 [71].

We observed that XPro1595 significantly reduced the levels of the pro-inflammatory cytokines IL-1β and IL-6 one day after SCI, but increased the anti-inflammatory cytokine IL-10 three days after SCI. The absence of IL-1β [72] and transient inhibition of IL-6 [73] are beneficial following SCI, while IL-10 significantly improves functional recovery and reduces SCI-induced TNF [18]. IL-1β, however, has been shown to be important for remyelination after cuprizone-induced demyelination [74]. Thus, one could speculate that the increased expression of IL-1β at 35 days after SCI, might enhance remyelination in the delayed phase after SCI, whereas IL-1β is more detrimental in the acute phase after SCI. These findings suggest that inhibition of solTNF for three consecutive days transiently favors an anti-inflammatory environment in the injured spinal cord the first days after SCI, which was sufficient to result in improved functional recovery. In support of this, we observed a significant reduction in the number of infiltrating macrophages and neutrophils in XPro1595-treated mice 14 days after SCI. Depletion of macrophages and neutrophils following SCI has been demonstrated to be beneficial after SCI [75], and evidence indicates they play an important role in adjusting neuroinflammation after SCI [76,77].

Microglia have recently been shown to play a pivotal role in shaping spinal cord repair after injury by orchestrating the response of other CNS-resident cells and the recruitment of leukocytes from the periphery [78]. Despite our findings of an altered inflammatory response following XPro1595 treatment, this did not affect the total number of microglia in the lesioned spinal cord or the temporal expression of typical microglial/macrophage markers such as *Arg1*, *Itgam*, *P2ry12*, *Trem2*, and *Cx3cr1*, except for the scavenger receptor *Cd68* that was significantly decreased in XPro1595-treated mice 7 days after SCI. These discrepancies may be due to experimental limitations related to the analysis of bulk spinal cord tissue, which may mask localized effects in specific regions or single-cell populations. To overcome these limitations, more focused analyses with higher spatial and cellular resolution are required. Indeed, our histological analysis focused on the peri-lesion area revealed a significant reduction in microglial Iba1^+^ cell average size and microglial CD68^+^ area fraction 21 days after SCI, suggesting decreased microglial reactivity in XPro1595-treated mice at this chronic time point. This is in line with previous studies performed in SCI [6,79] and other CNS injury models [80,81,82] showing that microglial reactive phenotype dynamically changes during lesion progression, shifting from early regenerative functions to a detrimental overstimulated profile at chronic stages. Therefore, the protective effects of XPro1595 may be related to the generation of a pro-regenerative environment in the spinal cord that translates into a reduction in microglial detrimental state in the chronic phase after SCI. This in turn could promote a decrease in lesion size, as previously observed [26], earlier wound healing, and enhanced subsequent remyelination after SCI.

It is well recognized that microglial phagocytosis is important during the acute lesion phase after SCI when the removal of damaged cells and myelin debris is critical for subsequent wound healing [83,84]. Conversely, at chronic disease stages, microglia overstimulation may result in uncontrolled and excessive phagocytosis of stressed-but-viable cells, termed ‘phagoptosis’, which contributes to disease worsening [85]. In our in vitro experimental model of LPS-stimulated primary microglia, which resembles acute rather than chronic injury stage, we observed increased phagocytosis following XPro1595 treatment, and impaired phagocytosis following etanercept treatment, in line with previous demonstrations of the importance of tmTNF-TNFR2 signaling for phagocytosis [60,65,86]. On this basis, enhanced microglial phagocytosis at early stages after injury may potentially contribute to improved functional outcomes in XPro1595-treated mice. This is in line with data showing that, in vivo, phagocytosis can be enhanced by reduced IL-6 levels, as temporal blocking of IL-6 promotes a pro-regenerative macrophage phenotype with increased phagocytosis after SCI [73]. Moreover, IL-10 is known to polarize microglia to become more pro-regenerative with increased phagocytosis, whereas the pro-inflammatory phenotype of microglia can be induced by TNF and has suppressed phagocytosis [87]. This may explain the changes observed in the spinal cord in the acute phase after SCI, resulting in decreased leukocyte infiltration. On the other hand, the finding of reduced microglial CD68 expression at late injury stage lines with dampened microglial phagoptosis and support improved chronic lesion recovery. It should be noted that phagocytic macrophages also are present in the lesioned spinal cord [37] and it is possible that XPro1595 treatment differently affects macrophage phagocytosis.

Activation of CNS-resident cells after SCI contributes to the infiltration of peripheral immune cells, and the interaction between these cells underlies the glial scar and chronic inflammation that together contribute to inhibiting the formation of axonal regeneration and remyelination. Furthermore, immediate therapy with a non-selective TNF antagonist supported axonal regeneration after peripheral nerve injury [88]. We therefore investigated whether XPro1595 treatment improved corticospinal fiber density/regrowth and PSD95 levels after SCI. However, we did not observe any apparent changes in fiber density, number of BDA^+^ cells, or PSD95 levels between treatment groups. In contrast, we found a significant increase in MBP expression in XPro1595-treated mice as compared to saline-treated mice, and MBP levels were associated with MBS scores. We previously demonstrated improved functional outcome and reduced secondary injury in SCI mice treated with XPro1595 [26], and a similar outcome with improved remyelination and axonal regeneration has been observed in XPro1595-treated myelin oligodendrocyte glycoprotein (MOG)-induced autoimmune encephalomyelitis mouse models [89]. In this study, inhibition of solTNF signaling provoked an increase or preservation in myelinated axons and likely a subsequent reduction of degenerated neurons. Furthermore, a study by Brambilla et al. [89] suggested that the reduced neuronal pathology, resulting in increased neuronal survival, was linked to tmTNF-TNFR2-initiated neuroprotective signaling. Although we did not observe any significant differences in TNFR2 levels, it is possible that selective inhibition of solTNF leads to increased tmTNF-TNFR2 signaling and improved locomotor functions, as we have previously demonstrated [26]. This is supported by the observed strong positive correlation between MBP levels and BMS scores.

Finally, as XPro1595 decreased the age-dependent increase in the overall number of CD4^+^ T cells in 5xFAD mice [58] and TNF has an important role in the regulation of T cells (specifically regulatory T-cells (T_reg_ cells) that are believed to be regulated through TNFR2 signaling [16]), we also analyzed the temporal expression of T cell markers, *Cd3*, *Cd8*, *Cd4*, and *Foxp3*, but observed no differences between treatment groups. This suggests that XPro1595 does not affect T-cell responses after SCI, which was further supported by comparable numbers of T cells between treatment groups.

## 5. Conclusions

Altogether, this study verifies our previous findings of improved functional recovery after SCI following topical XPro1595 treatment. We further demonstrate that epidural administration of XPro1595 for 3 consecutive days, is a feasible approach to transiently inhibit pro-inflammatory signaling, potentially affecting pro-regenerative processes after SCI. Our study consolidates XPro1595 as a promising new therapeutic treatment after SCI.

## Figures and Tables

**Figure 1 biology-12-00845-f001:**
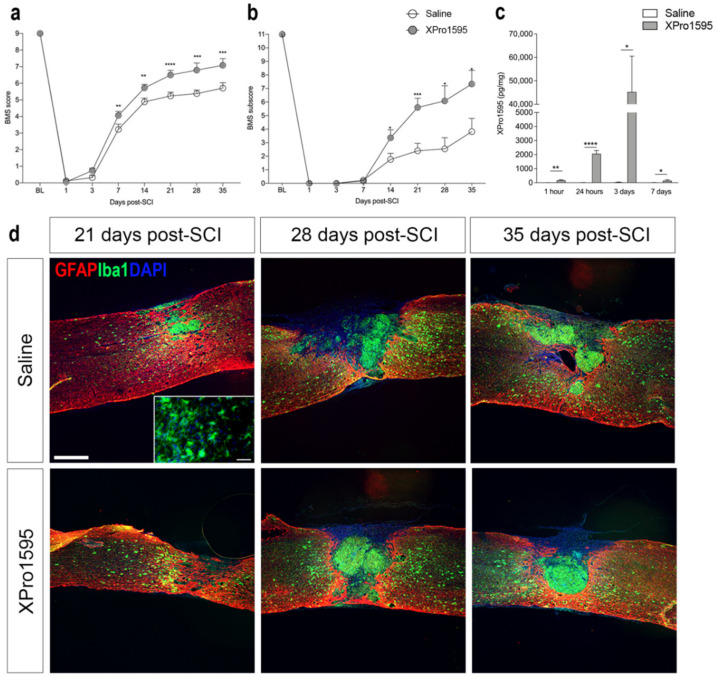
Topical inhibition of solTNF improves functional outcomes after SCI. (**a**) Evaluation of hindlimb locomotor function. XPro1595- and saline-treated mice were tested 1 and 3 days after SCI and weekly thereafter for 5 weeks. Motor behavior was scored under blinded conditions with the BMS (n_(day 1–21)_ = 25 mice/treatment group and n_(day 28–35)_ = 11–12 mice/treatment group). Two-way ANOVA (Time: F_3_._76,152_._7_ = 724.4, *p* < 0.0001, Treatment: F_1,48_ = 13.41, *p* = 0.0006, Interaction: F_7,284_ = 5.07, *p* < 0.0001) followed by Fisher’s LSD test: t(day 1)_332_ = 0.21, *p* = 0.36; t(day 3)_332_ = 1.50, *p* = 0.14; t(day 7)_332_ = 2.99, *p* = 0.003; t(day 14)_332_ = 2.99, *p* = 0.002; t(day 21)_332_ = 4.49, *p* = 0.00001; t(day 28)_332_ = 3.50, *p* = 0.0005; and t(day 35)_332_ = 3.40, *p* = 0.0008. (**b**) Evaluation of BMS subscores (n_(day 1–21)_ = 25 mice/treatment group and n_(day 28–35)_ = 11–12 mice/treatment group). Two-way ANOVA (Time: F_7,270_ = 122.8, *p* < 0.0001, Treatment: F_1,270_ = 32.53, *p* < 0.0001, Interaction: F_7,270_ = 5.26, *p* < 0.0001) followed by Fisher’s LSD test: t(day 1)_332_ = 0.21, *p* = 0.36; t(day 3)_332_ = 1.50, *p* = 0.14; t(day 7)_48_ = 0.44, *p* = 0.66; t(day 14)_48_ = 3.62, *p* = 0.04; t(day 21)_48_ = 3.62, *p* = 0.0007; t(day 28)_21_ = 2.50, *p* = 0.02; and t(day 35)_21_ = 2.49. *p* = 0.02. (**c**) XPro1595 levels 1 h as well as 1, 3, and 7 days after SCI measured by electrochemiluminescence in saline- and XPro1595-treated mice (n = 4–5 mice/treatment group). (**d**) Sections of spinal cords from saline- and XPro1595-treated mice with 21, 28, or 35 days survival after SCI double labeled for astroglial GFAP (red) and microglial/macrophage Iba1 (green). 4′,6-diamidino-2-phenylindole (DAPI) (blue) was used as a nuclear marker. Scale bar = 200 μm. Insert: Iba1^+^ ramified microglia located in the peri-lesion area of a saline-treated mouse 21 days after SCI. Scale bar = 40 μm. Data are presented as mean ± SEM, * *p* < 0.05, ** *p* < 0.01, *** *p* < 0.001, **** *p* < 0.0001. GFAP, glial fibrillary acidic protein; Iba1, ionized calcium-binding adaptor molecule 1; SCI, spinal cord injury.

**Figure 2 biology-12-00845-f002:**
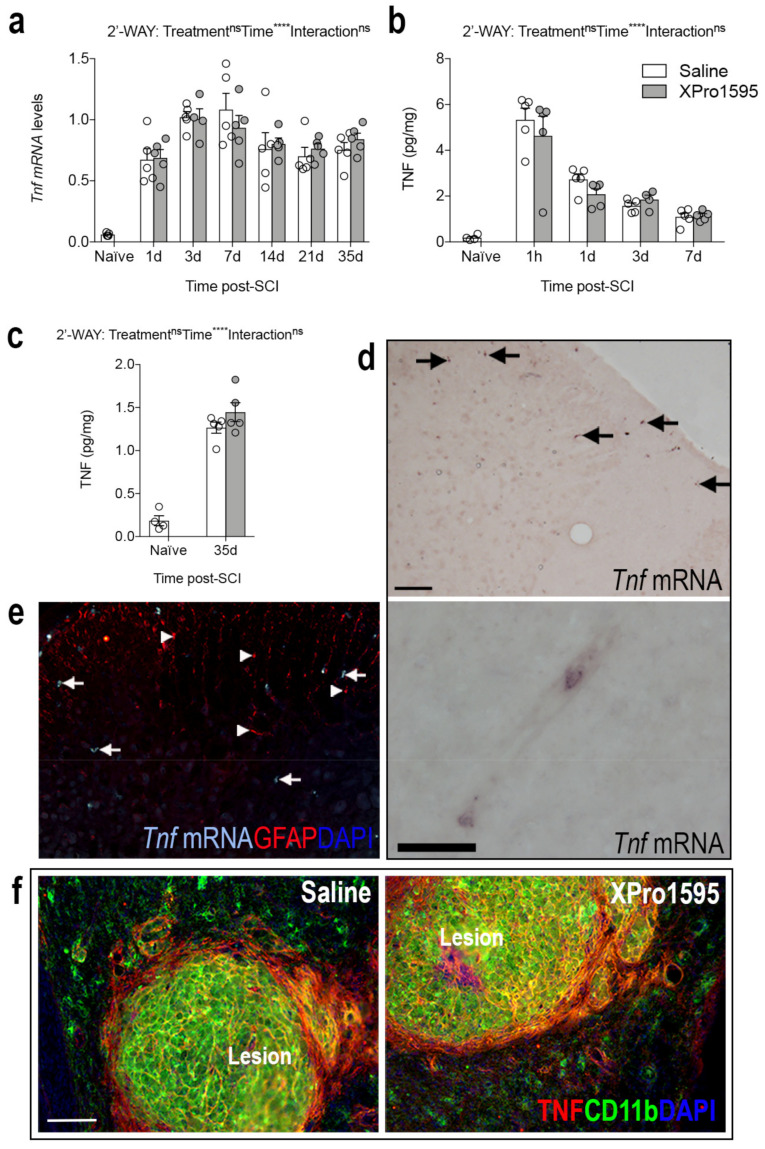
XPro1595 treatment does not affect the spatio-temporal expression of TNF after SCI. (**a**) Temporal expression of *Tnf* after SCI measured by real-time RT-qPCR (Time: F_6,55_ = 34.43, *p* < 0.0001, Treatment: F_1,55_ = 0.02, *p* = 0.88, Interaction: F_6,55_ = 0.50, *p* = 0.81). (**b**,**c**) TNF protein levels in the acute ((**b**); Time: F_5,44_ = 47.84, *p* < 0.0001; Treatment: F_1,44_ = 0.61, *p* = 0.44; Interaction: F_5,44_ = 0.76, *p* = 0.58) and delayed ((**c**); Time: F_1,14_ = 211, *p* < 0.0001, Treatment: F_1,14_ = 1.26, *p* = 0.28, Interaction: F_1,14_ = 1.26, *p* = 0.28) phases after SCI measured by chemiluminescence analysis. (**d**) *Tnf* mRNA^+^ cells in the posterior funiculi (arrows) and in a blood vessel of a mouse with 3 h survival after SCI. (**e**) Combined in situ hybridization for *Tnf* mRNA (turquoise, arrows) and immunofluorescence for astroglial GFAP (red, arrowheads) in the posterior funiculi 3 h after SCI, demonstrating the absence of co-localization. (**f**) Double immunofluorescent staining for TNF (red) and the microglial/macrophage marker CD11b (green) 35 days after SCI, demonstrating co-localization between TNF and CD11b^+^ cells within the lesion area in both saline- and XPro1595-treated mice. 4′,6-diamidino-2-phenylindole (DAPI) was used as a nuclear marker. Scale bars: (**d**,**e**) = 40 μm and (**f**) = 100 μm. Images in (**d**) are unpublished images of tissue sections acquired from parallel tissue sections from a previous study [6]. Data are presented as mean ± SEM with n = 5 mice/treatment group/time point. **** *p* < 0.0001. CD, cluster of differentiation; GFAP, glial fibrillary acidic protein; ns, not significant; SCI, spinal cord injury; TNF, tumor necrosis factor.

**Figure 3 biology-12-00845-f003:**
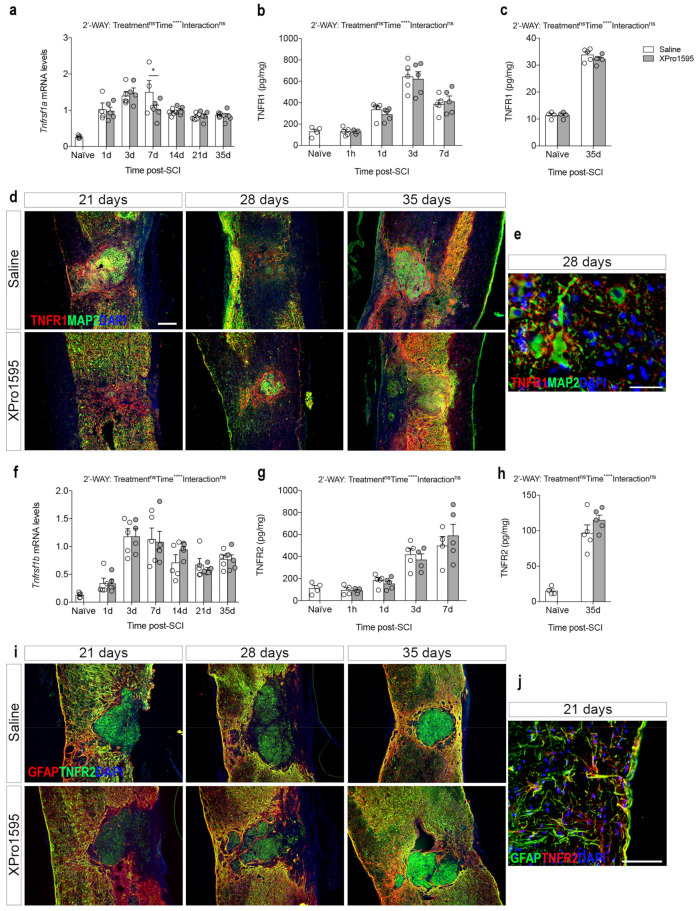
TNF receptor expression after SCI. (**a**) Temporal expression of *Tnfrsf1a* after SCI measured by real-time RT-qPCR analysis (Time: F_6,53_ = 24, *p* < 0.0001, Treatment: F_1,53_ = 1.00, *p* = 0.32, Interaction: F_6,53_ = 1.46, *p* = 0.21). (**b**,**c**) TNFR1 protein levels in the acute ((**b**), Time: F_3,32_ = 49.14, *p* < 0.0001, Treatment: F_1,32_ = 0.013, *p* = 0.72, Interaction: F_3,32_ = 0.25, *p* = 0.86) and delayed ((**c**), Time: F_1,14_ = 711.5, *p* < 0.0001, Treatment: F_1,14_ = 1.08, *p* = 0.32, Interaction: F_1,14_ = 1.08, *p* = 0.32) phases after SCI measured by chemiluminescence analysis. (**d**) Double immunofluorescent staining for TNFR1 (red) and neuronal MAP2 (green) 21, 28, and 35 days after SCI. (**e**) High magnification image of representative double immunofluorescent staining for TNFR1 (red) and MAP2 (green) in the peri-lesion area 28 days after SCI. (**f**) Temporal expression of *Tnfrsf1b* mRNA after SCI measured by real-time RT-qPCR analysis (Time: F_6,55_ = 23.09, *p* < 0.0001, Treatment: F_1,55_ = 0.04, *p* = 0.84, Interaction: F_6,55_ = 0.42, *p* = 0.87). (**g,h**) TNFR2 protein levels in the acute ((**f**), Time: F_3,29_ = 27.53, *p* < 0.0001, Treatment: F_1,29_ = 0.007, *p* = 0.93, Interaction: F_3,29_ = 0.61, *p* = 0.62) and delayed ((**g**), Time: F_1,14_ = 136.8, *p* < 0.0001, Treatment: F_1,14_ = 1.39, *p* = 0.26, Interaction: F_1,14_ = 1.39, *p* = 0.26) phases after SCI measured by chemiluminescence analysis. (**i**) Double immunofluorescent staining for TNFR2 (green) and astroglial GFAP (red) 21, 28, and 35 days after SCI. (**j**) High magnification image of representative double immunofluorescent staining for TNFR2 (red) and GFAP (green) in the peri-lesion area 21 days after SCI. 4′,6-diamidino-2-phenylindole (DAPI) was used as a nuclear marker. Scale bars: (**d**,**i**) = 100 µm and (**e**,**j**) = 40 µm. Data are presented as mean ± SEM with n = 5 mice/treatment group/time point. * *p* < 0.05, **** *p* < 0.0001. GFAP, glial fibrillary acidic protein; MAP2, microtubule-associated protein 2; ns, not significant; SCI, spinal cord injury; TNFR, tumor necrosis factor receptor.

**Figure 4 biology-12-00845-f004:**
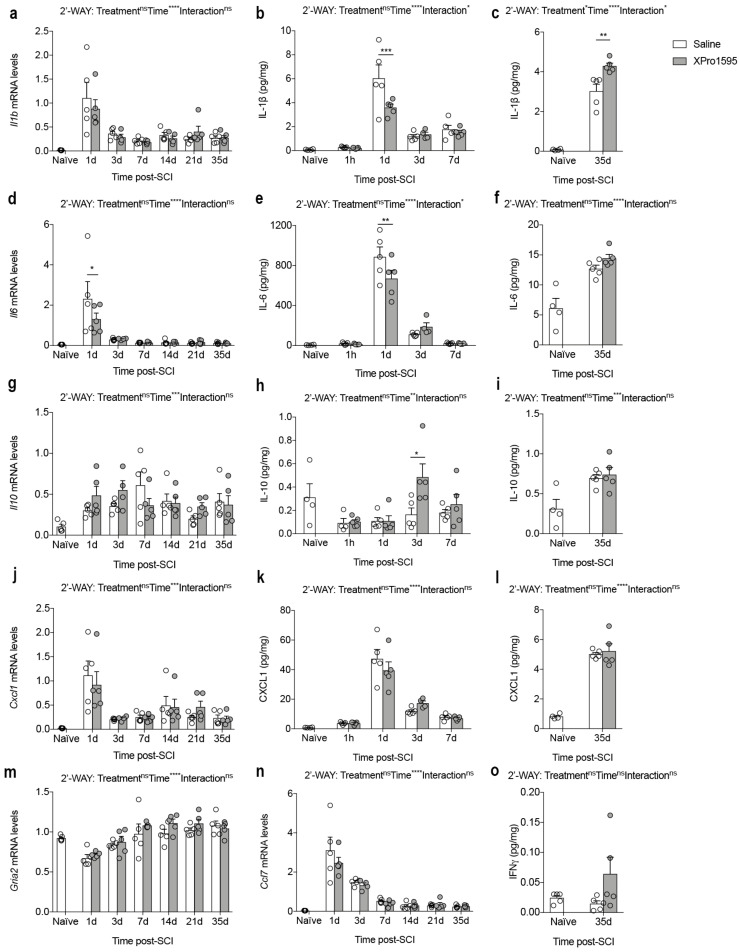
Inhibition of solTNF decreases the expression of pro-inflammatory and increases the expression of pro-regenerative markers after SCI. (**a**) Temporal expression of *Il1b* mRNA after SCI (Time: F_6,55_ = 14.98, *p* < 0.0001; Treatment: F_1,55_ = 0.33, *p* = 0.57; Interaction: F_6,55_ = 0.52, *p* = 0.79). (**b**,**c**) IL-1β protein levels in the acute ((**b**), Time: F_5,44_ = 46.78, *p* < 0.0001; Treatment: F_1,44_ = 1.37, *p* = 0.24; Interaction: F_5,44_ = 4.45, *p* = 0.002) and delayed ((**c**), Time: F_1,14_ = 282.7, *p* < 0.0001; Treatment: F_1,14_ = 8.82, *p* = 0.01; Interaction: F_1,14_ = 8.82, *p* = 0.01) phases after SCI. (**d**) Temporal expression of *Il6* mRNA after SCI (Time: F_6,55_ = 13.36, *p* < 0.0001; Treatment: F_1,55_ = 1.00, *p* = 0.32; Interaction: F_6,55_ = 1.23, *p* = 0.31). (**e**,**f**) IL-6 protein levels in the acute ((**e**), Time: F_5,44_ = 113.7, *p* < 0.0001; Treatment: F_1,44_ = 1.06, *p* = 0.31; Interaction: F_5,44_ = 2.97, *p* = 0.02) and delayed ((**f**), Time: F_1,14_ = 42.74, *p* < 0.0001; Treatment: F_1,14_ = 0.59, *p* = 0.45; Interaction: F_1,14_ = 0.59, *p* = 0.45) phases after SCI. (**g**) Temporal expression of *Il10* mRNA after SCI (Time: F_6,55_ = 4.76, *p* = 0.0006; Treatment: F_1,55_ = 0.49, *p* = 0.49; Interaction: F_6,55_ = 1.75, *p* = 0.13). (**h**,**i**) IL-10 protein levels in the acute ((**h**), Time: F_5,44_ = 16.48, *p* < 0.0001; Treatment: F_1,44_ = 3.12, *p* = 0.08; Interaction: F_5,44_ = 1.47, *p* = 0.22) and delayed ((**i**), Time: F_1,14_ = 19.45, *p* = 0.0006; Treatment: F_1,14_ = 0.06, *p* = 0.24; Interaction: F_1,14_ = 0.06, *p* = 0.81) phases after SCI. (**j**) Temporal expression of *Cxcl1* mRNA after SCI (Time: F_6,55_ = 11.27, *p* < 0.0001; Treatment: F_1,55_ = 0.01, *p* = 0.94; Interaction: F_6,55_ = 0.39, *p* = 0.88). (**k**,**l**) CXCL1 protein levels in the acute ((**k**), Time: F_5,44_ = 69.62, *p* < 0.0001; Treatment: F_1,44_ = 0.10, *p* = 0.76; Interaction: F_5,44_ = 1.21, *p* = 0.32) and delayed ((**l**), Time: F_1,14_ = 223.4, *p* < 0.0001; Treatment: F_1,14_ = 0.14, *p* = 0.71; Interaction: F_1,14_ = 0.14, *p* = 0.71) phases after SCI. (**m**) Temporal expression of *Gria2* mRNA after SCI (Time: F_5,56_ = 14.4, *p* < 0.0001; Treatment: F_1,56_ = 3.43, *p* = 0.07; Interaction: F_6,56_ = 0.65, *p* = 0.69). (**n**) Temporal expression of *Ccl7* mRNA after SCI (Time: F_6,55_ = 45.94, *p* < 0.0001; Treatment: F_1,55_ = 1.22, *p* = 0.27; Interaction: F_6,55_ = 0.71, *p* = 0.64). (**o**) IFNγ protein levels in the delayed phase after SCI (Time: F_1,9_ = 1.36, *p* = 0.27; Treatment: F_1,9_= 1.43, *p* = 0.26; Interaction: F_1,9_ = 1.43, *p* = 0.26). Gene expression is measured using real-time RT-qPCR and protein levels using electrochemiluminescence. Data are presented as mean ± SEM with n = 4–5/group/time point. * *p* < 0.05, ** *p* < 0.01, *** *p* < 0.001, **** *p* < 0.0001. CCL, CC chemokine ligand; CXCL, chemokine (C-X-C motif) ligand; Gria2, glutamate ionotropic receptor AMPA type subunit 2; IL, interleukin; IFNγ, interferon-gamma; ns, not significant; SCI, spinal cord injury.

**Figure 5 biology-12-00845-f005:**
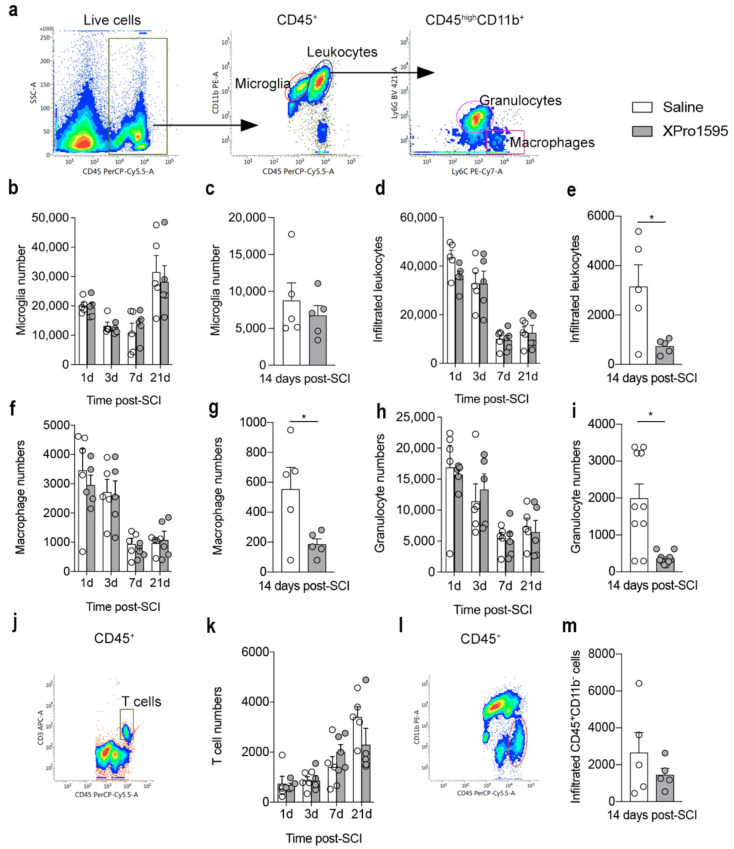
Inhibition of solTNF transiently decreases peripheral immune cell infiltration after SCI. (**a**) Dot blots showing the gating strategy for CD45^dim^CD11b^+^ microglia, CD45^high^CD11b^+^ leukocytes, CD45^high^CD11b^+^Ly6C^+^Ly6G^−^ macrophages, and CD45^high^CD11b^+^Ly6C^+^Ly6G^+^ granulocytes. (**b**,**c**) Changes in microglial numbers 1, 3, 7, and 21 days ((**b**), Interaction: F_3,32_ = 0.29, *p* = 0.83, Time: F_3,32_ = 13.40, *p* < 0.0001, Treatment: F_1,32_ = 0.09, *p* = 0.77) and 14 days ((**c**), t_8_ = 0.74, *p* = 0.48) after SCI. (**d**,**e**) Changes in the number of infiltrating leukocytes 1, 3, 7, and 21 days ((**d**), Interaction: F_3,32_ = 0.76, *p* = 0.53, Time: F_3,32_ = 43.89, *p* < 0.0001, Treatment: F_1,32_ = 0.96, *p* = 0.33) and 14 days ((**e**), t_8_ = 0.16, *p* = 0.88) after SCI. (**f**,**g**) Changes in the number of macrophages 1, 3, 7, and 21 days ((**f**), Interaction: F_3,32_ = 0.21, *p* = 0.89, Time: F_3,32_ = 18.08, *p* < 0.0001, Treatment: F_1,32_ = 0.45, *p* = 0.51) and 14 days ((**g**), t_8_ = 2.46, *p* = 0.04) after SCI. (**h**,**i**) Changes in the number of granulocytes 1, 3, 7, and 21 days ((**h**), Interaction: F_3,32_ = 0.22, *p* = 0.88, Time: F_3,32_ = 11.51, *p* < 0.0001, Treatment: F_1,32_ = 0.01, *p* = 0.93) and 14 days ((**i**), t_8_ = 2.77, *p* = 0.02) after SCI. (**j**) Dot blot showing the gating strategy for CD3^+^ T cells. (**k**) Changes in the number of CD3^+^ T cells 1, 3, 7, and 21 days after SCI (Interaction: F_3,32_ = 1.97, *p* = 0.14, Time: F_3,32_ = 15.13, *p* < 0.0001, Treatment: F_1,32_ = 0.30, *p* = 0.59). (**l**) Dot plot showing the gating strategy for CD45^high^CD11b^−^ lymphocytes. (**m**) Changes in the number of CD45^high^CD11b^−^ lymphocytes 14 days after SCI (t_8_ = 1.05, *p* = 0.32). Data are presented as mean ± SEM with n = 5/treatment group/time point. * *p* < 0.05. CD, cluster of differentiation; SCI, spinal cord injury.

**Figure 6 biology-12-00845-f006:**
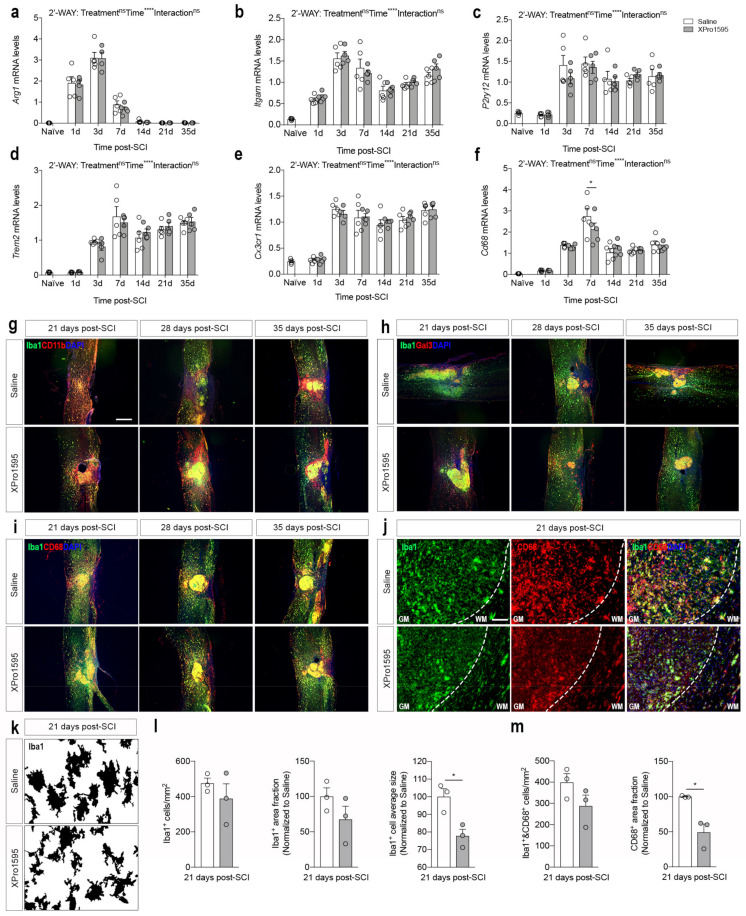
Inhibition of solTNF alters the morphology of CD68^+^Iba1^+^ cells after SCI. (**a**–**f**) Temporal expression of *Arg1* ((**a**), Interaction: F_6,55_ = 0.16, *p* = 0.99; Time: F_6,55_ = 135, *p* < 0.0001; Treatment: F_1,55_ = 0.45, *p* = 0.50), *Itgam* ((**b**), Interaction: F_6,55_ = 0.48, *p* = 0.82; Time: F_6,55_ = 155.29, *p* < 0.0001; Treatment: F_1,55_ = 0.63, *p* = 0.43), *P2ry12* ((**c**), Interaction: F_6,56_ = 0.72, *p* = 0.63; Time: F_6,56_ = 32.01, *p* < 0.0001; Treatment: F_1,56_ = 0.36, *p* = 0.55), *Trem2* ((**d**), Interaction: F_6,56_ = 0.56, *p* = 0.76; Time: F_6,56_ = 63.22, *p* < 0.0001; Treatment: F_1,56_ = 3.736, *p* = 1.00), *Cx3cr1* ((**e**), Interaction: F_6,55_ = 0.28, *p* = 0.94; Time: F_6,55_ = 81.79, *p* < 0.0001; Treatment: F_1,55_ = 0.09, *p* = 0.77), and *Cd68* ((**f**), Interaction: F_6,55_ = 1.38, *p* = 0.24; Time: F_6,55_ = 55.81, *p* < 0.0001; Treatment: F_1,55_ = 1.06, *p* = 0.31) mRNA levels after SCI (n = 5/treatment group/time point). (**g**–**i**) Immunofluorescent double labeling of Iba1^+^ cells (green) with CD11b ((**g**), red) Gal3 ((**h**), red), or CD68 ((**i**), red) at 21, 28, and 35 days after SCI (n = 2–3/group/time point). (**j**) Representative high-magnification images of Iba1^+^CD68^+^ cells located in the peri-lesion area used for morphological analyses 21 days after SCI. DAPI (blue) was used as a nuclear marker. (**k**) Representative converted greyscale images of Iba1^+^ microglia located in the peri-lesion area 21 days after SCI. (**l**,**m**) Morphological analysis of Iba1^+^ ((**l**), Iba1^+^ cells/mm^2^: t_4_ = 0.97, *p* = 0.39, Iba1^+^ area fraction: t_4_ = 1.46, *p* = 0.22, Iba1^+^ cell average size: t_4_ = 3.75, *p* = 0.02) and Iba1^+^CD68^+^ microglia (m, Iba1^+^&CD68^+^ cells/mm^2^: t_4_ = 1.68, *p* = 0.17, CD68^+^ area fraction: t_4_ = 4.30, *p* = 0.01) located 0–500 µm from the lesion border 21 days after SCI (3 sections per mouse, n = 3 mice/treatment group). Results are presented as mean ± SEM, * *p* < 0.05, **** *p* < 0.0001. Scale bars: (**g**–**i**) = 200 μm, (**j**) = 40 µm. Arg, arginase 1; CD, cluster of differentiation; Cx3cr1, CX3C motif chemokine receptor 1; Iba1, ionized calcium-binding adaptor molecule 1; Itgam, integrin subunit alpha M; Gal3, galectin-3; ns, not significant; P2ry12, purinergic receptor P2Y12; SCI, spinal cord injury; Trem2, triggering receptor expressed on myeloid cells 2.

**Figure 7 biology-12-00845-f007:**
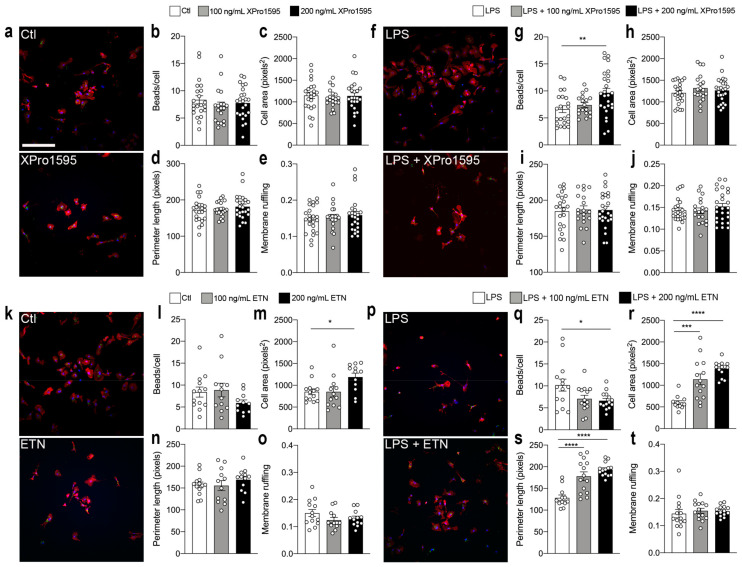
Selective inhibition of solTNF increases microglial phagocytosis. (**a**) Untreated control (Ctl) and XPro1595-treated primary microglial cultures stained for Iba1 (red) and DAPI (blue) demonstrating phagocytosis of FluoSpheres Carboxylate-Modified Microspheres (FSCMs, green). (**b**) Engulfed numbers of beads/cell in Ctl microglia (n = 22) and in 100 ng/mL (n = 19) and 200 ng/mL (n = 24) XPro1595-treated microglia (*p* = 0.51). (**c**–**e**) Morphological features of microglia under control conditions and after XPro1595 treatment; cell area ((**c**), *p* = 0.79), perimeter length ((**d**), *p* = 0.48) and membrane ruffling ((**e**), *p* = 0.69). (**f**) Representative images demonstrating phagocytosis of FSCMs in lipopolysaccharide (LPS)- (100 ng/mL) and LPS + XPro1595-treated microglia. (**g**) Engulfed number of beads/cell in LPS-treated microglia (n = 22) and in LPS + 100 ng/mL (n = 18) and LPS + 200 ng/mL (n = 28) XPro1595-treated microglia (*p* = 0.004). (**h**,**j**) Morphological features of microglia after LPS stimulation with and without XPro1595 treatment; cell area ((**h**), *p* = 0.46), perimeter length ((**i**), *p* = 0.95), and membrane ruffling ((**j**), *p* = 0.48). (**k**) Representative images demonstrating phagocytosis of FSCMs in Ctl and etanercept (ETN)-treated microglia. (**l**) Engulfed number of beads/cell in Ctl microglia (n = 13) and in 100 ng/mL (n = 12) and 200 ng/mL (n = 11) ETN-treated microglia (*p* = 0.21). (**m**–**o**) Morphological features of microglia under Ctl conditions and after ETN treatment; cell area ((**m**), *p* = 0.03), perimeter length ((**n**), *p* = 0.49), and membrane ruffling ((**o**), *p* = 0.22). (**p**) Representative images demonstrating phagocytosis of FSCMs in LPS- and LPS + ETN-treated microglia. (**q**) Engulfed number of beads/cell in LPS-treated microglia (n = 14) and in LPS + 100 ng/mL (n = 14) and LPS + 200 ng/mL (n = 14) ETN-treated microglia (*p* = 0.03). (**r**–**t**) Morphological features of microglia after LPS stimulation with and without ETN treatment; cell area ((**r**), *p* = 0.0003), perimeter length ((**s**), *p* < 0.0001), and membrane ruffling ((**t**), *p* = 0.76). Scale bar = 80 μm. Data are presented as mean ± SEM, individual data represent the number (n) of coverslips/technical replicates from 3 independent experiments. * *p* < 0.05, ** *p* < 0.01, *** *p* < 0.001, **** *p* < 0.0001. Ctl, control; ETN, etanercept; LPS, lipopolysaccharide.

**Figure 8 biology-12-00845-f008:**
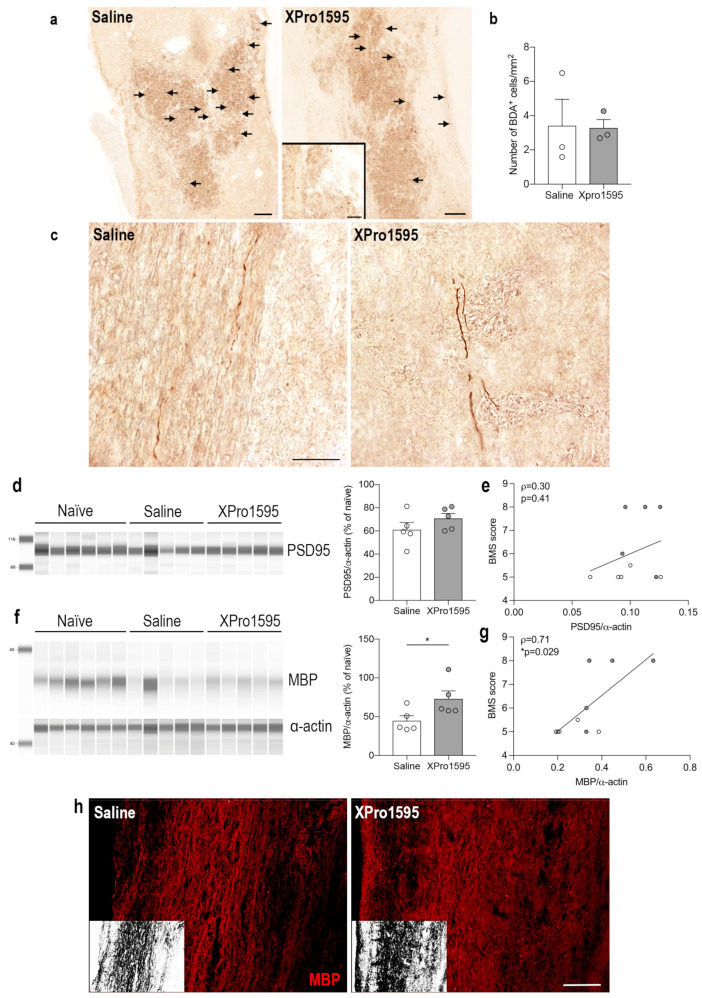
Selective inhibition of solTNF preserves myelin but does not alter the density of biotinylated dextran amine (BDA)^+^ cells and neuronal plasticity after SCI. (**a**) Representative images demonstrating corticospinal tract BDA^+^ axonal debris (arrows) engulfed by phagocytic-like cells located within the lesion area 24 days after SCI. (**b**) Estimation of the number of BDA^+^ cells/mm^2^ in the lesion area (n = 3/treatment group). (**c**) Representative images demonstrating BDA^+^ corticospinal axons with anterogradely transported BDA rostral to the lesion 24 days after SCI. (**d**) Quantification of post-synaptic PSD95 35 days after SCI. (**e**) Correlation analysis of PSD95 levels and motor function (BMS) 35 days after SCI. (**f**) Quantification of MBP 35 days after SCI. (**g**) Correlation analysis of MBP levels and motor function (BMS) 35 days after SCI. Protein data are normalized to α-actin expression and presented as a percentage of naïve mice (n = 6) with n = 5 mice/treatment group. White circles represent saline-treated mice and grey circles represent XPro1595-treated mice. * *p* < 0.05. (**h**) Representative images of MBP-labeled myelin structures in the peri-lesion area 35 days after SCI. Inserts show binary masks of MBP staining. Scale bars: (**a**,**h**) = 100 µm, and insert in (**a**,**c**) = 40 µm. Data are presented as mean ± SEM. MBP, myelin basic protein; PSD95, postsynaptic density protein 95; SCI, spinal cord injury.

**Figure 9 biology-12-00845-f009:**
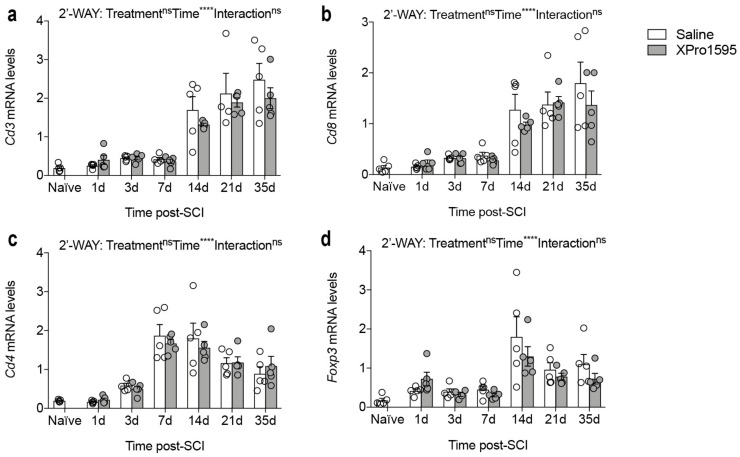
Temporal gene expression of phenotypic T-cell markers after SCI. (**a**–**d**) Temporal expression of *Cd3* ((**a**), Interaction: F_6,56_ = 0.61, *p* = 0.72; Time: F_6,56_ = 35.32, *p* < 0.0001; Treatment: F_1,56_ = 1.59, *p* = 0.21), *Cd8* ((**b**), Interaction: F_6,56_ = 0.57, *p* = 0.75; Time: F_6,56_ = 24.77, *p* < 0.0001; Treatment: F_1,56_ = 1.18, *p* = 0.28), *Cd4* ((**c**), Interaction: F_6,56_ = 0.38, *p* = 0.88; Time: F_6,56_ = 28.41, *p* < 0.0001; Treatment: F_1,56_ = 0.15, *p* = 0.70), and *Foxp3* ((**d**), Interaction: F_6,55_ = 0.93, *p* = 0.48; Time: F_6,55_ = 11.85, *p* < 0.0001; Treatment: F_1,55_ = 1.73, *p* = 0.19) mRNA levels after SCI. Data are presented as mean ± SEM with n = 4–5/group. **** *p* < 0.0001. CD, cluster differentiation; Foxp3, forkhead box 3; ns, not significant; SCI, spinal cord injury.

**Table 1 biology-12-00845-t001:** Real-time RT-qPCR primers.

*GENE*	PRIMER SEQUENCE (5′-3′)	ACCESSION NO.	ANNEALING TEMP. (TA)	PRODUCT (TM)	NO. OF CYCLES
** *TNF* **	F- AGGCACTCCCCCAAAAGATG	NM_001278601.1	60 °C	81.5 °C	40
	R- TCACCCCGAAGTTCAGTAGACAGA				
** *TNFRSF1A* **	F- GCCCGAAGTCTACTCCATCATTTG	NM_011609.4	60 °C	80.5 °C	40
	R- GGCTGGGGAGGGGGCTGGAGTTAG				
** *TNFRSF1B* **	F- GCCCAGCCAAACTCCAAGCATC	NM_011610.3	60 °C	78.5 °C	40
	R- TCCTAACATCAGCAGACCCAGTG				
** *IL1B* **	F- TGCCACCTTTTGACAGTGATG	NM_008361.4	60 °C	77 °C	40
	R- CAAAGGTTTGGAAGCAGCCC				
** *IL6* **	F- AGGATACCACTCCCAACAGA	NM_001314054.1	60 °C	77.5 °C	40
	R- ACTCCAGGTAGCTATGGTACTC				
** *IL10* **	F- CCAGGTGAAGACTTTCTTTCAAAC	NM_010548.2	61.5 °C	81.5 °C	45
	R- AGTCCAGCAGACTCAATACACAC				
** *CXCL1* **	F- GCTGGGATTCACCTCAAGAAC	NM_008176.3	60 °C	80 °C	40
	R- TGTGGCTATGACTTCGGTTTG				
** *ITGAM* **	F- GCCTGTCACACTGAGCAGAA	NM_008401.2	55 °C	79.5 °C	40
	R- TGCAACAGAGCAGTTCAGCA				
** *CX3CR1* **	F- TCCCATCTGCTCAGGACCTC	NM_009987.4	55 °C	77.5 °C	40
	R- GGCCTCAGCAGAATCGTCAT				
** *TREM2* **	F- TGCTGGAGATCTCTGGGTCC	NM_031254.3	60 °C	80.5 °C	40
	R- AGGTCTCTTGATTCCTGGAGGT				
** *ARG1* **	F- ATGAAGAGCTGGCTGGTGTG	NM_007482.3	60 °C	81 °C	40
	R- CCAACTGCCAGACTGTGGTC				
** *P2RY12* **	F- GCCAGTGTCATTTGCTGTCAC	NM_027571.4	60 °C	81.5 °C	40
	R- TAGATGCCACCCCTTGCACT				
** *CCL7* **	F- GATCTCTGCCACGCTTCTGT	NM_013654.3	62 °C	82 °C	45
	R- GCATTGGGCCCATCTGGTT				
** *CD68* **	F- GGTGGAAGAAAGGCTTGGGG	NM_001291058.1	60 °C	78.5 °C	45
	R- GAGACAGGTGGGGATGGGTA				
** *CD8* **	F- ACAACTGCCCCAACCAAGAA	NM_009858.3	60 °C	79.5 °C	40
	R- TGCATGTCAGGCCCTTCTG				
** *CD4* **	F- CCAACAGCGCCAGGCA	NM_013488.3	60 °C	81.0 °C	40
	R- CTCTTCTGCATCCGGTGGG				
** *FOXP3* **	F- GCGAAAGTGGCAGAGAGGTAT	NM_001199347.1	60 °C	81.5 °C	45
	R- AAGTTGCCGGGAGCTGGAG				
** *CD3* **	F- GAGGATGCGGTGGAACACTT	NM_007648.5	60 °C	81.5 °C	45
	R- ATGTTCTCGGCATCGTCCTG				
** *GRIA2* **	F- AGCAAGGCGTCTTAGACAAGC	NM_001083806.3	60 °C	78.0 °C	40
	R- GGGCACTGGTCTTTTCCTTACT				
** *HPRT1* **	F- TCCTCAGACCGCTTTTTGCC	NM_013556.2	60 °C	84 °C	40
	R- TCATCATCGCTAATCACGACGC				

## Data Availability

Requests to access datasets should be directed to klabertsen@health.sdu.dk.

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
