# Peer review of "Selective Inhibition of Soluble Tumor Necrosis Factor Alters the Neuroinflammatory Response following Moderate Spinal Cord Injury in Mice"

_biology, 2023, doi:10.3390/biology12060845_

Round 1
Reviewer 1 Report
In their paper "Selective inhibition of soluble tumor necrosis factor alters the 2 neuroinflammatory response following moderate spinal cord 3 injury in mice" describe Lund and co-workers present the treatment of mice with XPro1595. This compound, further only mentioned as XPro, is a pegylated dimeric form of TNF. Which, due to its affinity to trimerize, disrupts intact soluble TNF thereby shifting the overall systemic response to tmTNF and therefore to TNFR2. The authors apply this compound to a mouse model of SCI. They can demonstrate reduced myelin lesions and improved clinical outcomes and some highly significant alterations in the immune response. The authors show also that TNFR levels did not change significantly. This is an important information and most likely will lead to discussions in the field. Despite the great effort the authors made in analyzing changes in the immune response not only in cytokine levels, macrophages/microglia but also granulocytes, leukocytes and T cells, it would have been of some advantage to have a look at subpopulations of T cells especially if the authors would see some altered infiltration of T regs in XPro treated animals with respect to PBS treated controls. With respect to microglia and macrophages if they can see more specific subtypes (e.g. CD68 or others). Knowing that in some cases performing these experiments would be out of scope as they would require additional experiments it would be highly interesting and could potentially improve this already excellent study.
Author Response
Response to reviewer #1: We thank the reviewer for the very positive review of our manuscript and for the constructive comments. We agree that analysis of T cell responses in our SCI is of interest and that this could be done using flow cytometry as well. We do, however, believe that given the low number of infiltrated T cells at all time points (<4000 cells on average), looking for subpopulations of T cells such as Treg will be very difficult using flow cytometry. We hope that the reviewer appreciates that we attempted to look for potential changes in T cell responses using qPCR analysis (Figure 9). We did, however, not observe any significant differences between treatment groups in Cd3, Cd4, Cd8, or Foxp3 mRNA expression levels and therefore decided not to proceed with a detailed T cell analysis, as this would have required additional mice as well as new experiments.
With respect to CD68 analyses, flow cytometric analysis of CD68+ microglia and macrophages would also require additional animals and experiments, which is, as also mentioned by the reviewer, out of the scope of this paper. As demonstrated in Figure 6, Cd68 was the only microglia/macrophage gene that changed between treatment groups. We therefore decided to look for changes in CD68+ microglia located in the peri-infarct area, as CD68+ macrophages are too densely packed within the lesion core to allow for proper analysis of these cells. We agree with the reviewer that future studies focusing more on specific microglial and macrophage subtypes are highly interesting.
Reviewer 2 Report
In the work “Selective inhibition of soluble tumor necrosis factor alters the neuroinflammatory response following moderate spinal cord injury in mice”, Lund et al. builds upon previous findings that central application of the inhibitor of soluble TNF, XPro1595, leads to functional recovery following SCI by the decrease of the lesion size and inflammatory response. In the current work they are able to examine gene and protein changes in pro and anti-inflammatory molecules in response to XPro1595-treatment in the acute and chronic phases of SCI, which are transient and minimal. Further they show that there is a transient decrease at 14dpi of leukocytes in the lesion site with XPro1595-treatment. 7dpi there is a transient decrease in Cd68, a general marker of activated phagocytic microglia/macrophages. This is coupled with a decrease in Cd68 area fraction 21dpi, coupled with decreased Iba1+ cell size and morphology. Further, XPro1595 increases microglial phagocytosis compared to the non-specific TNF inhibitor etanercept following LPS. MBP levels were increased by XPro1595 treatment and correlated with higher BMS scores. A great deal of work was done to examine the mechanistic benefits provided by XPro1595 treatment following SCI. The English is with minimal typos and grammatically correct without need for revision. Although the new findings are not striking they do add to the current knowledge and should be published with the following changes addressed.
Abstract:
The abstract while accurate might slightly oversell the findings as it is not clear that these changes were not permanent but mostly transient at various timepoints.
Introduction:
The introduction laid out the premise of the reasoning to pursue the work but it would be helpful to clearly outline here and in the results of which work was a repetition of previous findings and which differ.
Methods:
Please add animal authority authorization as you have done in previous work.
Were the micro-osmotic pumps that delivered treatment for 3 days removed afterwards? Typically mini-pumps, especially on mice, are removed after a two week period. If not please mention this in the methods as well.
For replication work it would be good to mention the catheter type used as some drugs stick to certain catheters and cannot be released and there is no need to withhold this information.
Although listed in the methods about how many animals/group there is at each timepoint it is a bit hard to follow. It would be useful to add into the graphs the individual points in addition to the bars which has become the new standard for such graphs that also allows the reader to see the numbers as well as the variation.
When was the in-situ hybridization done? It isn’t listed.
Was Hprt1 not changing at the various timepoints as it was used as a control?
Why was a different method used for BMS than Two-way ANOVA? Maybe I am not aware of the reasoning of why this would be a better method.
Results:
For all graphs it would be useful (as mentioned above) that the individual values are shown with dots within the bar graphs along with the mean and error bars. This will help the reader understand how many animals are in each group and the true variation observed.
Figure 1 is basically a confirmation of the last work, however it might be worthwhile to either include the lesion analysis here as you did previously or mention the previous finding.
In general in this section I had difficulty always knowing what had been done before or the reasoning of what was being done. It may just be a preference but it would help the reader with understanding the flow and logic if some of the sections in the discussion were moved here. Not necessary, but a suggestion.
Figure 2 is not clear if there has been shown previously that inhibition of solTNF protein feedback to affecting mRNA levels of TNF or leads to TNF degradation and this is why this experiment was undertaken. The spatial location does make sense but it unclear to me if this is confirming work done before or completely novel work. Having looked up the review from these authors it appears to confirming previous work. What would have been more interesting is if XPro1595 changes the cellular distribution. F) appears to be novel and the most interesting but I would have even preferred higher magnification pictures to know if the localization within the cells was altered and not just that it was found in and around the lesion site with macrophage and microglia.
Figure 3, again it is unclear that using XPro1595 that forms inactive heterodimers with solTNF would lead to changes in the gene expression of Tnfsrf1a or Tnfsrf1b or protein changes in TNFR1 or TNFR2, is there a known link solTNF levels following and the TNF receptors expression? D) A bit confused by the MAP2 staining, why does it look similar to the TNFR2 in the lesion site? Is it not a very specific antibody or maybe mislabeled (the core of the lesion shouldn’t be filled with neurons)? Also, why is MAP2 used here and GFAP later in the figure? Why do the lesion sites go from bigger to smaller to bigger overtime with saline? F) does not replicate the increase seen at 7dpi with XPro1595 with previous work. H) the TNFR2 levels at 21 dpi with XPro1595 appear less but at 35dpi appear more, is this the case?
Figure 4 IL6 findings seem to be novel, although it is unclear why it is only at 1 dpi.
Figure 5c,e,g and i representing 14dpi separately from Figure 5 b,d,f and h timeline? And why is the values seem to be off in 14dpi compared to 7 and 21dpi?
Figure 6 adds to previous Western work showing Iba1 increases with SCI at lesion site and decreased with XPro1595 treatment, that also the Iba1+ cell average cell size decreases along with CD68+ area fraction. Again this was from studying the previous work.
Figure 7’s primary microglial phagocytosis response to LPS with XPro1595 treatment was a novel take to address the effects that XPro1595 has on microglia activity. However, it is unclear if the individual dots represent individual cells? It should represent biological replicates done in technical replicates. Figure 7r, why is the cell area LPS numbers so much lower than that of h) or even c) and m), that may account for the difference than a true increase observed.
In Figure 8, given the data, why wasn’t dieback axonal analysis from the lesion site done? This would give us information that should be in the samples shown here. In addition, staining for sprouting of 5HT fibers could also be contributing to functional recovery and would be of interest to examine without extra experiments performed, just additional stainings. Figure 8f replicates work done previously, however g) appears to have inflated values from what I view in f), why would this be? G) does appear to be new information.
Figure 9 in new data although there are no differences seen with XPro1595 treatment.
Discussion:
I find the last sentence of the first paragraph a bit of an over statement:
Our findings demonstrate that topical XPro1595 treatment after SCI modifies the inflammatory response to favor a pro-regenerative environment, resulting in myelin preservation, and improved functional outcome.
There does not seem to be any demonstration of regeneration within this work and further it is unclear if this preservation of myelin or remyelination.
On line 908, referencing #70 regarding Glut2R (Gria2) changes were similar to the work here, but ref 70 shows Gria1 differences which would be easy enough to examine here, why wasn’t it?
In the following paragraph “These findings suggest that inhibition of solTNF for three consecutive days favors an anti-inflammatory environment in the injured spinal cord the first days after SCI, leading to improved functional recovery.”, were only shown at 14dpi and not 21dip so how exactly does this lead to the functional recovery that starts as early as 7dpi and lasts until 35dpi?
Line 975 and 977 has typos.
Last two sentences seem to be an overstatement again: We further demonstrate that epidural administration of XPro1595 for 3 consecutive days, is an efficient approach to inhibit pro-inflammatory signaling and boost pro-regenerative processes after SCI. Our study consolidates XPro1595 as a promising new therapeutic treatment after SCI.
It is clear that a great deal of work and time went into this study and it should be published. With these edits, clarifications and additions the work will provide some answers to the mechanisms underlying the functional recovery provided by XPro1595 following SCI.
Author Response
Reviewer #2
In the work “Selective inhibition of soluble tumor necrosis factor alters the neuroinflammatory response following moderate spinal cord injury in mice”, Lund et al. builds upon previous findings that central application of the inhibitor of soluble TNF, XPro1595, leads to functional recovery following SCI by the decrease of the lesion size and inflammatory response. In the current work they are able to examine gene and protein changes in pro and anti-inflammatory molecules in response to XPro1595-treatment in the acute and chronic phases of SCI, which are transient and minimal. Further they show that there is a transient decrease at 14dpi of leukocytes in the lesion site with XPro1595-treatment. 7dpi there is a transient decrease in Cd68, a general marker of activated phagocytic microglia/macrophages. This is coupled with a decrease in Cd68 area fraction 21dpi, coupled with decreased Iba1+ cell size and morphology. Further, XPro1595 increases microglial phagocytosis compared to the non-specific TNF inhibitor etanercept following LPS. MBP levels were increased by XPro1595 treatment and correlated with higher BMS scores. A great deal of work was done to examine the mechanistic benefits provided by XPro1595 treatment following SCI. The English is with minimal typos and grammatically correct without need for revision. Although the new findings are not striking they do add to the current knowledge and should be published with the following changes addressed.
We thank the reviewer for the constructive comments made to our manuscript. We have addressed the points of criticisms to the best of our knowledge and our response is detailed below.
Abstract:
Reviewer 2, comment #1: The abstract while accurate might slightly oversell the findings as it is not clear that these changes were not permanent but mostly transient at various timepoints.
Response to #1: We agree with reviewer and included in the abstract that our findings are transient. We hope that this is clearer now.
Introduction:
Reviewer 2, comment #2: The introduction laid out the premise of the reasoning to pursue the work but it would be helpful to clearly outline here and in the results of which work was a repetition of previous findings and which differ.
Response to #2: We slightly modified the introduction to accommodate the reviewer’s request: “We previously demonstrated that selective inhibition of solTNF using the dominant-negative inhibitor XPro1595 reduced lesion volume and improved functional outcomes in mice subjected to SCI, probably by targeting microglial responses [26]. As we did not previously extensively study temporal neuroinflammatory responses, in the present work, we extend our observations by investigating the potential of selective solTNF inhibition in inflammatory and pro-regenerative processes after SCI.”, page 3, lines 95-97.
Methods:
Reviewer 2, comment #3: Please add animal authority authorization as you have done in previous work.
Response to #3: As requested by the journal, the animal authority authorization is listed under the paragraph Institutional Review Board Statement, where we included the following statement: “The animal study protocol was approved by the Danish Veterinary and Food Administration (J number 2013-15-2934-00924 and 2019-15-0201-01615) and experiments are reported in accordance with the ARRIVE guidelines. All efforts were made to minimize the animals’ pain and distress.”
Reviewer 2, comment #4: Were the micro-osmotic pumps that delivered treatment for 3 days removed afterwards? Typically mini-pumps, especially on mice, are removed after a two week period. If not please mention this in the methods as well.
Response to #4: The mini-osmotic pumps were not removed after surgery. As requested by the reviewer, we clarified this in the manuscript in the Materials and Methods section, page 4, lines 128-129.
Reviewer 2, comment #5: For replication work it would be good to mention the catheter type used as some drugs stick to certain catheters and cannot be released and there is no need to withhold this information.
Response to #5: As requested by the reviewer, we added the catheter type used in the study under the Materials and Methods section, page 4, lines 124-125: “The pumps were placed in such a way that the delivering end of the catheter (Micro-Renathane Tubing MRE-040, AgnTho’s AB, Lindingö, Sweden) was on top of the injured spinal cord.”
Reviewer 2, comment #6: Although listed in the methods about how many animals/group there is at each timepoint it is a bit hard to follow. It would be useful to add into the graphs the individual points in addition to the bars which has become the new standard for such graphs that also allows the reader to see the numbers as well as the variation.
Response to #6: As requested by the reviewer, we added into the graphs the individual points, when meaningful. We have changed the following graphs: Figure 2a-c, Figure 3a-c, Figure 3e-g, Figure 4a-o, Figure 5b-I, Figure 5k, Figure 5m, Figure 6a-f, Figure 6l-m, Figure 8b, Figure 8d, Figure 8f, and Figure 9a-d, as well as all graphs in the Supplemental figures.
Reviewer 2, comment #7: When was the in-situ hybridization done? It isn’t listed.
Response to #7: We clarified the time for the in situ hybridization in the materials and methods section, page 5, lines 162-164. The in situ hybridization was done 3 hours after SCI, when Tnf mRNA levels are known to peak in our model (Lund et al., Biology 2022).
Reviewer 2, comment #8: Was Hprt1 not changing at the various timepoints as it was used as a control?
Response to #8: Ct values for Hprt1 was consistent around 19 throughout the study in both saline- and XPro1595-treated mice. The average Ct values for each group at each of the time points are listed below.
|
|
Naive |
1D |
3D |
7D |
14D |
21D |
35D |
|
Saline |
19.05±0.41 |
18.94±0.22 |
19.70±0.32 |
19.43±0.37 |
18.90±0.26 |
19.38±0.16 |
19.36±0.23 |
|
XPro1595 |
|
18.95±0.26 |
19.68±0.33 |
19.06±0.18 |
19.10±0.25 |
19.36±0.26 |
19.35±0.24 |
We included the following sentence in the Material and Methods sections on page 6, lines 258-259: “The temporal cycle threshold (Ct) values for Hprt1 were found to be stable around 19 cycles and were comparable between treatment groups.”
Reviewer 2, comment #9: Why was a different method used for BMS than Two-way ANOVA? Maybe I am not aware of the reasoning of why this would be a better method.
Response to #9: The statistics was done using a two-way ANOVA followed by multiple t-test analysis. As we wanted to compare the two groups at each time point and were not interested in following the same group over time. We included the results of the two-way ANOVA in the figure legend for Figure 1.
Results:
Reviewer 2, comment #10: For all graphs it would be useful (as mentioned above) that the individual values are shown with dots within the bar graphs along with the mean and error bars. This will help the reader understand how many animals are in each group and the true variation observed.
Response to #10: Please see reply to #6 above.
Reviewer 2, comment #11: Figure 1 is basically a confirmation of the last work, however it might be worthwhile to either include the lesion analysis here as you did previously or mention the previous finding.
Response to #11: We agree that Figure 1 is a confirmation of our previous work. This was to certify that we could indeed repeat our previous findings of improved functional outcome after SCI in XPro1595-treated mice (Novrup et al., J Neuroinflam 2014). In the present study, we extend our studies to investigate the effect of solTNF inhibition on the post-SCI inflammatory responses. As we and others (see e.g., Basso et al., J Neurotrauma, 2006), have extensively demonstrated that BMS scores as a measure of functional outcome correlates with white matter sparring/lesion size, we prioritized to use tissue for analysis of neuroinflammatory responses. We modified the paragraph in the materials and methods section to the following “As we previously demonstrated that topical XPro1595 treatment for three consecutive days significantly reduced lesion volumes and improved functional outcome in mice [26], we initially wanted to confirm the therapeutic efficacy of XPro1595 in our model”, pages 10-11, lines 429-431.
Reviewer 2, comment #12: In general in this section I had difficulty always knowing what had been done before or the reasoning of what was being done. It may just be a preference but it would help the reader with understanding the flow and logic if some of the sections in the discussion were moved here. Not necessary, but a suggestion.
Response to #12: We thank the reviewer for this comment, however, according to the guidelines given in journal’s template, we have kept the original format of a Results section, followed by a general Discussion and a Conclusion.
Reviewer 2, comment #13: Figure 2 is not clear if there has been shown previously that inhibition of solTNF protein feedback to affecting mRNA levels of TNF or leads to TNF degradation and this is why this experiment was undertaken. The spatial location does make sense but it unclear to me if this is confirming work done before or completely novel work. Having looked up the review from these authors it appears to confirming previous work. What would have been more interesting is if XPro1595 changes the cellular distribution. F) appears to be novel and the most interesting but I would have even preferred higher magnification pictures to know if the localization within the cells was altered and not just that it was found in and around the lesion site with macrophage and microglia.
Response to #13: The reviewer is correct, TNF is known to act in an autocrine manner and we included a sentence and a reference to clarify this this in section 3.2: As Tnf is known to act in an autocrine manner both at the transcriptional and protein level (Raffaele 2020, Cells), we wanted to assess whether XPro1595 treatment influenced the temporal expression levels of Tnf mRNA and TNF protein after SCI, we per-formed real-time RT-qPCR and electrochemiluminescence analyses.”, page 12, lines 469-470. The spatial and cellular localization of TNF was done to i) confirm previous findings of TNF expression in our SCI model and to ii) to whether XPro1595 affected this. To accommodate the reviewer’s point, we included the following sentence “Then we also wanted to confirm the cellular and spatial localization of TNF in our model [49],.”, page 10, lines 475-476. We agree with reviewer that it would be interesting to investigate changes in cellular localization of TNF, however this is out of the scope of this paper, as this requires new animals and experiments.
Reviewer 2, comment #14: Figure 3, again it is unclear that using XPro1595 that forms inactive heterodimers with solTNF would lead to changes in the gene expression of Tnfsrf1a or Tnfsrf1b or protein changes in TNFR1 or TNFR2, is there a known link solTNF levels following and the TNF receptors expression? D) A bit confused by the MAP2 staining, why does it look similar to the TNFR2 in the lesion site? Is it not a very specific antibody or maybe mislabeled (the core of the lesion shouldn’t be filled with neurons)? Also, why is MAP2 used here and GFAP later in the figure? Why do the lesion sites go from bigger to smaller to bigger overtime with saline? F) does not replicate the increase seen at 7dpi with XPro1595 with previous work. H) the TNFR2 levels at 21 dpi with XPro1595 appear less but at 35dpi appear more, is this the case?
Response to #14: XPro1595 increased TNFR2 levels 7 days after SCI as compared to etanercept and saline in the study by Novrup et al., J Neuroinflam 2014, and we wanted to investigate the temporal changes in TNFR2 in the present study. We included the following sentence in the manuscript: “We previously showed that the protective effects of inhibiting pro-inflammatory solTNF function were associated with upregulation of neuroprotective TNFR2 expression in the spinal cord [26]”, page 14, lines 505-507.
In the lesion core, there is a mix of dying cells taken up by infiltrating macrophages. Therefore, MAP2+ staining will be found inside phagocytosing macrophages.
MAP2 is expressed mostly by neurons and TNFR2 is expressed mostly by astrocytes, therefore co-localization studies were done using these markers. We elaborated on this in the results section page 14, line 518.
It is true that figure 3f does not replicate what we previously found 7 days after SCI. In our previous work (Novrup et al., 2014), we measured TNFR2 using WB whereas in the present study, we used electrochemiluminescence. Therefore, there could be differences in the techniques used and, there is a greater variability in the present data, which may account for this. This was already discussed in the discussion of the manuscript.
As we in the Figure 3h show qualitative images to demonstrate the co-localization, you cannot draw any conclusion on the expression levels. We did not attempt to draw any conclusions on quantitative levels for that reason.
Reviewer 2, comment #15: Figure 4 IL6 findings seem to be novel, although it is unclear why it is only at 1 dpi.
Response to #15: We agree with the reviewer that this is interesting. The temporal profile of IL-6 in SCI is that it increases transiently 1 day after SCI whereafter it goes back to baseline levels. We therefore would not expect any effect at later time points.
Reviewer 2, comment #16: Figure 5c,e,g and i representing 14dpi separately from Figure 5 b,d,f and h timeline? And why is the values seem to be off in 14dpi compared to 7 and 21dpi?
Response to #16: Flow at 14 dpi was done using a different protocol and different fluorophores that do not allow direct comparisons. We therefore found it more correct to show data in two separate figures. We included at sentence clarifying this in the materials and methods section: “Samples collected at day 14 was run using the same protocol but without the CD3 marker and using different fluorophore combinations and therefore are presented separately.”, page 10, lines393-394..
Reviewer 2, comment #17: Figure 6 adds to previous Western work showing Iba1 increases with SCI at lesion site and decreased with XPro1595 treatment, that also the Iba1+ cell average cell size decreases along with CD68+ area fraction. Again this was from studying the previous work.
Response to #17: The reviewer is correct; we extend the findings of changed Iba1 levels in XPro1595 treated mice to more cellular analyses. We included a short sentence on this: “We previously demonstrated that XPRo1595 affects Iba1 protein levels in the spinal cord [26].”, page 20, lines 685-686.
Reviewer 2, comment #18: Figure 7’s primary microglial phagocytosis response to LPS with XPro1595 treatment was a novel take to address the effects that XPro1595 has on microglia activity. However, it is unclear if the individual dots represent individual cells? It should represent biological replicates done in technical replicates. Figure 7r, why is the cell area LPS numbers so much lower than that of h) or even c) and m), that may account for the difference than a true increase observed.
Response to #18: The dots represent individual culture wells and can therefore be considered a single biological replicate. We therefore decided to keep the figure as is. We clarified this in the legend for Figure 7. As we performed separate experiments, panel h and panel t cannot be compared.
Reviewer 2, comment #19: In Figure 8, given the data, why wasn’t dieback axonal analysis from the lesion site done? This would give us information that should be in the samples shown here. In addition, staining for sprouting of 5HT fibers could also be contributing to functional recovery and would be of interest to examine without extra experiments performed, just additional stainings. Figure 8f replicates work done previously, however g) appears to have inflated values from what I view in f), why would this be? G) does appear to be new information.
Response to #19: We agree with the reviewer that investigating axonal dieback from the lesion site as well sprouting of 5HT fibers, however, this experiment is outside the scope of this manuscript as this would require new experiments with mice. Regarding figure f, the bar graphs represent changes in relation to naïve, while g) represents the actual gene changes.
Reviewer 2, comment #20: Figure 9 in new data although there are no differences seen with XPro1595 treatment.
Response to #20: Yes, we agree that these are new data and unfortunately, there are no differences.
Discussion:
Reviewer 2, comment #21: I find the last sentence of the first paragraph a bit of an over statement:
Our findings demonstrate that topical XPro1595 treatment after SCI modifies the inflammatory response to favor a pro-regenerative environment, resulting in myelin preservation, and improved functional outcome.
There does not seem to be any demonstration of regeneration within this work and further it is unclear if this preservation of myelin or remyelination.
Response to #21: We agree with the reviewer and changed the sentence: “Our findings suggest that topical XPro1595 treatment after SCI transiently modifies the inflammatory response to favor a protective environment, resulting in increased myelin integrity, and improved functional outcome.”, page 27, lines 851-854.
Reviewer 2, comment #22: On line 908, referencing #70 regarding Glut2R (Gria2) changes were similar to the work here, but ref 70 shows Gria1 differences which would be easy enough to examine here, why wasn’t it?
Response to #22: We agree with the reviewer that we could also have tested Gria1, however as TNF has been shown to trigger rapid membrane insertion of AMPA receptors and, in some cases, specific insertion of GluR2-lacking, Ca2+ permeable AMPA receptors into motor neurons, enhancing their susceptibility to slow excitotoxic injury, we decided only to investigate Gria2.
Reviewer 2, comment #23: In the following paragraph “These findings suggest that inhibition of solTNF for three consecutive days favors an anti-inflammatory environment in the injured spinal cord the first days after SCI, leading to improved functional recovery.”, were only shown at 14dpi and not 21dip so how exactly does this lead to the functional recovery that starts as early as 7dpi and lasts until 35dpi?
Response to #23: We agree with the reviewer that this sentence was not clear and have now modified the sentence: “These findings suggest that inhibition of solTNF for three consecutive days transiently favors an anti-inflammatory environment in the injured spinal cord the first days after SCI, which was sufficient to result in improved functional recovery.”, page 28, lines 902-905.
Reviewer 2, comment #24: Line 975 and 977 has typos.
Response to #24: Thank you for pointing this out. We checked and corrected the typos.
Reviewer 2, comment #25: Last two sentences seem to be an overstatement again: We further demonstrate that epidural administration of XPro1595 for 3 consecutive days, is an efficient approach to inhibit pro-inflammatory signaling and boost pro-regenerative processes after SCI. Our study consolidates XPro1595 as a promising new therapeutic treatment after SCI.
Response to #25: We agree with the reviewer and modified the sentence accordingly: “We further demonstrate that epidural administration of XPro1595 for 3 consecutive days, is a feasible approach to transiently inhibit pro-inflammatory signaling, potentially affecting pro-regenerative processes after SCI.”, page 29, lines 985-988.
It is clear that a great deal of work and time went into this study and it should be published. With these edits, clarifications and additions the work will provide some answers to the mechanisms underlying the functional recovery provided by XPro1595 following SCI.
Reviewer 3 Report
The authors present an interesting study on the effect of a specific inhibitor of the soluble form of TNF (XPro1595). The results presented show the interest of XPro1595 both on functional recovery and on tissue modulation. The authors also demonstrate that this treatment makes it possible to regulate the inflammatory response by acting in particular on the circulating cells and not on the resident cells. Although the results are particularly interesting, I still have some comments and questions about this work.
1. In Figure 1d, we have the feeling that the meninges are sometimes torn following SCI (28 days in particular) whereas in other photos this does not seem to be the case. The authors should clarify whether this parameter was taken into account and whether animals on the basis of meningeal integrity were included or excluded from the analysis of the results. Indeed, it is conceivable that the rupture of the meninges plays a role in the invasion of cells from the general circulation.
2. In Figures 1 and 3, the authors should add the PDGFrB marker to highlight the fibrotic component of the lesion and also measure the effect of XPro1595 treatment on this parameter. In particular, this would make it possible to see whether the fibroblasts express TNFR2 in Figure 3.
3. In Figure 3, the authors should provide "zoomed in" images so that the co-stainings can be checked.
4. In figure 4C, I do not understand how IL1B expression can be higher at D35 compared to D7, the authors should discuss this.
5. In figure 6H, the spinal cord images are not put in the same way.
6. Figure 8, the authors should add to the MBP WB immunostaining and quantification on this parameter.
7. Optional: Why did the authors want to analyse the effects of XPro1595 treatment on microglial cells and not on macrophage lines, when the results show an effect of this treatment on circulating cells and not on resident cells? If possible, the authors should perform these experiments on macrophage cell lines or at least discuss about it.
Author Response
Reviewer #3:
The authors present an interesting study on the effect of a specific inhibitor of the soluble form of TNF (XPro1595). The results presented show the interest of XPro1595 both on functional recovery and on tissue modulation. The authors also demonstrate that this treatment makes it possible to regulate the inflammatory response by acting in particular on the circulating cells and not on the resident cells. Although the results are particularly interesting, I still have some comments and questions about this work.
We thank the reviewer for the positive review of our manuscript and the constructive comments/questions that we have tried to address to the best of our knowledge below.
Reviewer 3, comment #1: In Figure 1d, we have the feeling that the meninges are sometimes torn following SCI (28 days in particular) whereas in other photos this does not seem to be the case. The authors should clarify whether this parameter was taken into account and whether animals on the basis of meningeal integrity were included or excluded from the analysis of the results. Indeed, it is conceivable that the rupture of the meninges plays a role in the invasion of cells from the general circulation.
Response to #1: The moderate SCI contusion model does not lead to rupture of the meninges in our hands. The tearing of the meninges observed in some figures is a result from removing the spinal cord from the spinal canal, when the mice are sacrificed, which is inevitable. Meningeal integrity, therefore, is intact in all mice during the entire experiment.
Reviewer 3, comment #2: In Figures 1 and 3, the authors should add the PDGFrB marker to highlight the fibrotic component of the lesion and also measure the effect of XPro1595 treatment on this parameter. In particular, this would make it possible to see whether the fibroblasts express TNFR2 in Figure 3.
Response to #2: As requested by the reviewer, we tried to stain for PDGFRb in our spinal cord tissue. Unfortunately, in our hands, immunofluorescence for PDGFRb was unsuccessful. However, we agree that TNFR2 labeling, at the time points analyzed, appears mostly localized within the scar tissue, which is formed by a combination of glial cells (GFAP+) as well as fibroblasts. We thus modified our conclusions accordingly (see page 14, lines 527-530).
Reviewer 3, comment #3: In Figure 3, the authors should provide "zoomed in" images so that the co-stainings can be checked.
Response to #3: As requested by the reviewer, we have included “zoomed in” images for Figure 3d – 28 days saline and Figure 3h – 28 days XPro1595.
Reviewer 3, comment #4: In figure 4C, I do not understand how IL1B expression can be higher at D35 compared to D7, the authors should discuss this.
Response to #4: As stated in the Materials and Methods section, page 8, lines 284-286: “Analysis of tissue derived from mice with survival time 1 and 24 hours and 3 and 7 days after SCI was performed separately from mice with 35 days survival after SCI, and therefore analyzed as two separate experiments”, day 35 was done separately from the rest of the experiments. The V-Plex and Ultra PLEX kits used for the two experiments were not from the same lot and had the kits had changed between the two experiments. The catalogue numbers of the different kits are already listed in the Materials and Methods, page 8. It was therefore not possible to directly compare day 35 to the other time points. We included the naïve mice for comparison.
Reviewer 3, comment #5: In figure 6H, the spinal cord images are not put in the same way.
Response to #5: We acknowledge that the spinal cord images are not put in the same way in some of the images in Figure 6H. This is due to the way the tissue sections from these mice were placed on the slides during cryostat processing. We can, unfortunately, not rotate the images without compromising the magnification of the images.
Reviewer 3, comment #6: Figure 8, the authors should add to the MBP WB immunostaining and quantification on this parameter.
Response to #6: As requested, we provide representative immunostaining panels showing higher MBP integrity in the peri-lesion area of XPro1595-treated mice vs saline (see Figure 8h). However, we believe that WB, when applicable, represents a much more reliable method to quantify protein levels than immunofluorescence. We therefore decided to not include the quantification of MBP immunostaining.
Reviewer 3, comment #7: Optional: Why did the authors want to analyse the effects of XPro1595 treatment on microglial cells and not on macrophage lines, when the results show an effect of this treatment on circulating cells and not on resident cells? If possible, the authors should perform these experiments on macrophage cell lines or at least discuss about it.
Response to #7: We thank the reviewer for this comment, which we think indeed is a valid point. We did, however, in the present study focus our analyses on microglial reactions mainly. Future studies will help us study the effect of XPro1595 in macrophages. As requested by the reviewer, we discuss this in the Discussion section, page 29 lines 250-252.
Round 2
Reviewer 2 Report
I appreciate your detailed answers to my inquiries and the changes that were done. Overall my concerns were addressed except for the following.
Comment #9: You are interested in the group differences over time and that is why you performed the two-way ANOVA, showing he interaction. Typically for the posthoc analysis you mentioned, you would use a Fisher's LSD and not multiple t-tests.
Comment #14: Thank you for the clarification, but I still find the qualitative work lacking when the colocalization with magnifications are difficult to see, aren't provided for each panel and are unclear which part of the picture they come from (dotted box around the area helps). Also, the scale bar is listed as one instead of two given there is a magnification. Even if that magnification is done artificially, it requires better panels than what is shown here.
Comment #18: No, different coverslips do not equal biological replicates. These are primary cells and each isolation is a biological replicate. Please change this to technical replicates. However, going back and doing biological replicates would be the proper way to do this experiment.
Author Response
Please find response to reviewer #2:
Reviewer comment #1:
Comment #9: You are interested in the group differences over time and that is why you performed the two-way ANOVA, showing he interaction. Typically for the posthoc analysis you mentioned, you would use a Fisher's LSD and not multiple t-tests.
Reply to #1: The Fisher’s LSD test works just like the t test with one exception. The t test computes a pooled standard deviation from the two groups being compared while the Fisher LSD test uses the pooled standard deviation. In the present study, we did assume that all samples (entire table) are from populations with the same SD. We changed the this in the statistical analyses to accommodate the reviewer’s request and changed multiple t test to Fisher’s LSD test.
Reviewer comment #2:
Comment #14: Thank you for the clarification, but I still find the qualitative work lacking when the colocalization with magnifications are difficult to see, aren't provided for each panel and are unclear which part of the picture they come from (dotted box around the area helps). Also, the scale bar is listed as one instead of two given there is a magnification. Even if that magnification is done artificially, it requires better panels than what is shown here.
Reply to #2: We included a high magnification (40x) image of a TNFR1 and MAP2 staining 28 days after SCI and a TNFR2 and GFAP staining 21 days after SCI to accommodate the reviewer’s request. These images are representative for both groups and at all time points. Please note, that we in the new Figure 3, image j, we had to change the fluorophores of the secondary antibodies so that GFAP is detected with a 488-conjugated primary antibody and the TNFR2 is detected by a 594-conjugated antibody.
Reviewer comment #3:
Comment #18: No, different coverslips do not equal biological replicates. These are primary cells and each isolation is a biological replicate. Please change this to technical replicates. However, going back and doing biological replicates would be the proper way to do this experiment.
Reply to #3: As requested by the reviewer, we changed “biological replicate” to “technical replicate”.